# Improving Generative Adversarial Networks via Adversarial Learning in Latent Space

## Abstract

Generative Adversarial Networks (GANs) have been widely studied as generative models, which map a latent distribution to the target distribution. Although many efforts have been made in terms of backbone architecture design, loss function, and training techniques, few results have been obtained on how the sampling in latent space can affect the final performance, and existing works on latent space mainly focus on controllability. We observe that, as the neural generator is a continuous function, two close samples in latent space would be mapped into two nearby images, while their quality can differ much as the quality is not a continuous function in pixel space. From the above continuous mapping function perspective, on the other hand, two distant latent samples are also possible to be mapped into two close images. If the latent samples are mapped in aggregation into limited modes or even a single mode, mode collapse occurs. Accordingly, we propose adding an implicit latent transform before the mapping function to improve latent $z$ from its initial distribution, e.g., Gaussian. In this paper, this is specifically achieved by using the iterative fast gradient sign method (I-FGSM). We further propose new GAN training strategies to obtain better generation mappings w.r.t quality and diversity by introducing targeted latent transforms into the bi-level optimization of GAN. Experimental results on visual data show that our method can effectively achieve improvement in both quality and diversity.

## 1 Introduction

Generative Adversarial Networks (GANs) (Goodfellow et al., 2014a) have shown effectiveness for generating high-fidelity data, especially for images under various settings (Ledig et al., 2017; Wang et al., 2018; Zhu et al., 2017; Zhan et al., 2019; Chakraborty et al., 2018). Based on the zero-sum game, the model is trained by the adversarial process between the generator and the discriminator. Many efforts have been made to achieve a more realistic generation from different perspectives. WGAN (Arjovsky et al., 2017), SNGAN (Miyato et al., 2018), LSGAN (Mao et al., 2017) aim to design better objective functions. ImprovedGAN (Salimans et al., 2016), AC-GAN (Odena et al., 2017) propose practical training techniques. While more complex network structures are studied in BigGAN (Brock et al., 2018), StyleGAN (Karras et al., 2019a), SAGAN (Zhang et al., 2019).

Despite the above progress, an intriguing and relatively less studied question is how the latent space affects the generation in terms of quality and diversity, which can be orthogonal to the above frequently studied factors like model architectures and training techniques. Specifically, we argue that the latent samples from a standard continuous distribution (e.g., Gaussian) can often be mapped to varying-quality samples for the generation. One reason is that the generator is a continuous function (by neural nets) while the generation quality (e.g., images in pixel space) is not. Moreover, it has also been shown in GAN literature (Khayatkhoei et al., 2018) that many image objects lie upon multiple disjoint manifolds, and a continuous mapping from continuous latent distribution, e.g., Gaussian must generate many invalid samples in the between of the manifolds when the generator seeks to cover all the meaningful modes. See Fig. 1 (a) for the schematic diagram. This fact also implies a dilemma: for complex real-world data, there is a trade-off between quality and diversity when the latent vectors are sampled from a simple uni-mode distribution, e.g., Gaussian, and the generator is a continuous mapping based on neural networks. Moreover, by realizing that the mapping function in fact allows for many-to-one mapping, mode collapse (Srivastava et al., 2017) can naturally happen when different latent samples are mapped into limited or even a single mode in target space.

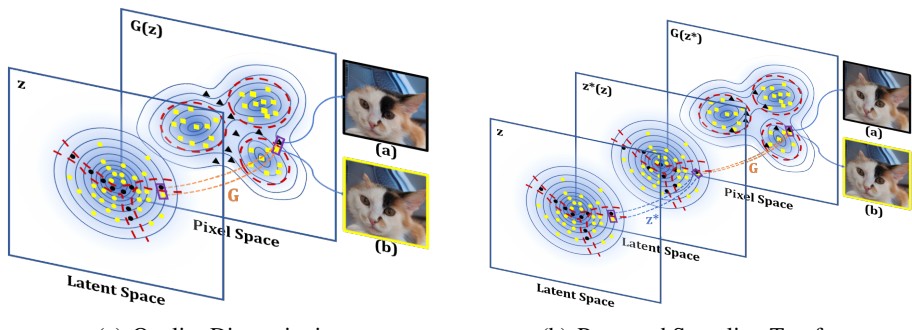

(a) Quality Discontinuity        (b) Proposed Sampling Tranform

Figure 1: Red dashed lines denote real data manifolds in pixel space and their inverse images in latent space; yellow circles and squares denote samples for sound generation in latent space and pixel space; black circles and squares denote samples for bad generation in latent space and pixel space; the orange dashed lines denote the generator mapping; the blue dashed lines denote $z$ transform in latent space; blue shades and solid blue lines denote the latent distribution and the generated distribution. Due to the continuity of the neural network-based generator, close samples in latent space will be mapped to close images in pixel space, while the images' quality can vary. Cat (a) and Cat (b) are close in pixel space, but Cat (a) loses its right ear. Our method allows bad latent $\mathbf{z}$ to transform towards the inverse image of the nearest real manifold, thus avoiding terrible generation.

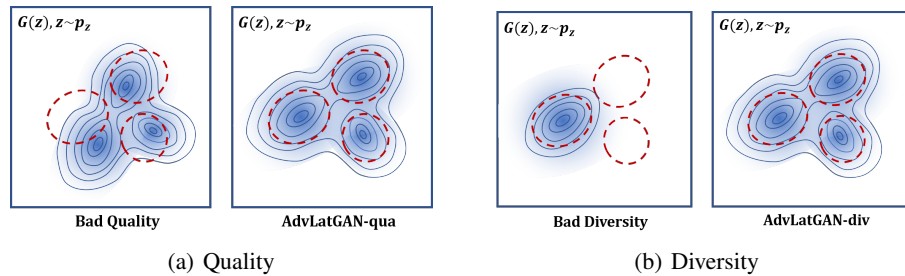

(a) Quality        (b) Diversity

Figure 2: Red dashed lines denote real data manifolds; blue shades and blue lines denote the generated distribution. The generated distributions mapped from a fixed latent distribution reflect the mapping quality. (a) shows the case of bad quality when the generated distribution does not match the real distribution well. (b) shows the case of bad diversity when the mapping misses modes. Our method tries to train mappings with better properties, as shown on the right side of each case.

The above contradiction between the discontinuity in generation data quality (e.g., images) and the continuity of latent distribution as well as mapping function, can be one of the significant factors to the well-known difficulty in GANs training, especially for generators, as one unreasonably hopes to enforce a discontinuous mapping via continuous neural nets. Meanwhile, the mapping property can largely affect the generative performance for functions with continuous nature, which is associated with most previous works devoted to improving generative performance.

To tackle these issues, we impose an extra (implicit) transform function $z^*(\cdot)$ on the raw sampling $\mathbf{z}$, and the generation can be written by $G(z^*(\mathbf{z}))$. From this perspective, most existing works, including the vanilla GAN, employ $z^*(\cdot)$ with an identical form. Specifically, the implicit function based transform $z^*(\cdot)$ can be achieved by updating the latent $\mathbf{z} \sim p(\mathbf{z})$ to minimize $loss_G$ (i.e. $\log(1 - D(G(\mathbf{z})))$) using iterative fast gradient sign method (I-FGSM) (Kurakin et al., 2016) with the parameters of $G$ and $D$ fixed, and we name the method as AdvLatGAN-z. See Fig. 1 (b) for the schematic diagram. On top of the first effort, we propose a novel training strategy to improve the generative mapping $G$ respectively for quality and diversity. See Fig. 2 for the schematic diagram. This is achieved by introducing $\mathbf{z}$ iterations (i.e., targeted implicit transforms on $\mathbf{z}$) into GAN training, specifically, updating $\mathbf{z} \sim p(\mathbf{z})$ using I-FGSM to find more optimization-friendly ones in the bi-level optimization of vanilla GAN. Respectively for quality and diversity, we use two iterative updating strategies for $\mathbf{z}$ under different objective functions in training, and we name the two training algorithms respectively as AdvLatGAN-qua and AdvLatGAN-div. **The highlights of this paper include:**

1) We rethink the role of the generator as a continuous function, which may incur quality discontinuity in target space and mode aggregation (collapse) when the latent samples are directly sampled from

a continuous distribution e.g. Gaussian. Based on above theoretical considerations, we propose a composite generation perspective by formulating the mapping from $G(\mathbf{z})$ to $G(z^*(\mathbf{z}))$. This treatment provides a new methodological guidance to the progress of the generative model.

2) We propose a new pipeline for both $G(\cdot)$ and $z^*(\cdot)$, by first training generative mappings for better generation quality and diversity, respectively, and then the transform $z^*(\cdot)$ is enforced and computed implicitly to obtain an improved distribution for more effective generation. Specifically, the training involves adversarial learning under different objective functions for quality and diversity, leading to new GAN models: AdvLatGAN-qua/div and their further variants.

3) Experimental results on synthetic and visual data show the notable improvement of our approach on both generation quality and diversity. Our technique is orthogonal to existing GAN techniques.

## 2 RELATED WORK

Implicitly adapting the raw latent vectors $\mathbf{z}$ from Gaussian to improve the generation relates to latent space learning for GAN. While the specific training algorithm using latent vector adapting to mine optimization-friendly samples in training relates to adversarial training, which views the latent adapting in training as an attack to mine hard samples.

**Latent space exploration in GANs.** Techniques have been proposed for latent space exploration for GANs. By utilizing the steerability of GANs in latent space, StyleGAN (Karras et al., 2019a) introduce an latent space transform by using a mapping network to better decouple the various styles and Jahanian et al. (2019) shift the distribution to fit camera movement transformation. InterFaceGAN (Shen et al., 2020) decouples entangled semantics with subspace projection to control facial attributes more precisely. Mishra et al. (2020), Mukherjee et al. (2019) and Wu et al. (2019) introduce the latent exploration into training to achieve clustering objectives or benefit GAN training. In addition to the preceding works, some recent works focus on improving latent sampling quality to generate high-quality images. For example, interactive evolutionary computation is used to make the generation process more controllable in Bontrager et al. (2018), which meanwhile also brings additional manual efforts. To reduce manual intervention, Roziere et al. (2020; 2021) use Koncept512 (Hosu et al., 2020) as a criterion to improve the quality of fake images. However, as discussed in Roziere et al. (2020), this strategy is somehow biased, can not work well in each dataset (e.g., Pokemons). The reason is that the quality estimator is not even designated to access the real-data distribution information. Instead, the generation is guided by a predefined score function.

Compared to the above works, we take a relevant idea to Roziere et al. (2020) that also adapt the latent vectors to an improved condition before input to the generator $G$. Differently, our latent adaption approach incorporates the target distribution and utilize I-FGSM iterations to adapt latent vectors.

**Mitigating mode collapse for GANs.** There are also efforts in mitigating mode collapse, which are based on the thinking that many real-world samples lie in disjoint manifolds (Khayatkhoei et al., 2018). One way is to introduce a collection of generators such that each generator may cover a specific mode since a single generator can hardly fulfill a discontinuous function (Khayatkhoei et al., 2018). However, more works focus on improving the properties of mapping, explicitly obtaining a mapping covering more modes (Elfeki et al., 2019; Meulemeester et al., 2020; Mao et al., 2019). Note that the generated distribution obtained by this method will contain invalid parts due to the continuity of the latent distribution and the mapping. This calls for selective sampling of the continuous latent distribution or directly amending the latent distribution, which is rarely noticed in these works.

**Adversarial samples and adversarial training.** Adversarial samples are malicious samples that can easily fool deep models, which are indistinguishable to human eyes. Since the seminal work (Szegedy et al., 2013), various adversarial attack methods have been devised (Goodfellow et al., 2014b; Akhtar & Mian, 2018; Kurakin et al., 2016; Madry et al., 2017). To defend the models from attacks, adversarial training is meanwhile developed (Goodfellow et al., 2014b; Wang et al., 2019; Nøkland, 2015). In particular, it is worth noting that several recent studies have shown that adversarial training can bring many benefits to GANs, not just improving robustness. RGAN (Zhang et al., 2020) shows that adversarial training can enhance the generalization of the generator and discriminator. Based on both theoretical analysis and empirical results, Zhong et al. (2020); Liu et al. (2021); Liu & Hsieh (2019) show that updating discriminator's parameters with the loss calculated by adversarial samples generated by authentic images can stabilize training and accelerate convergence.

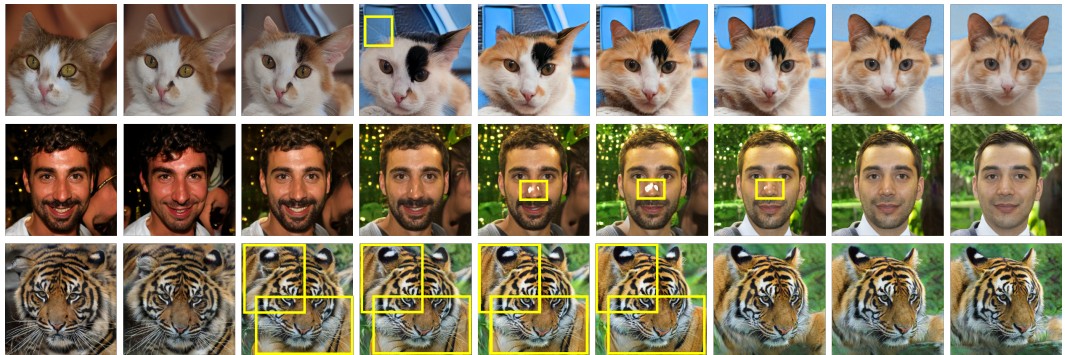

Figure 3: Generation examples of StyleGAN2-ada (Karras et al., 2020). Sampling equidistantly from $p(\mathbf{z})$ in the latent space, the samples in the generated distribution $p_g(\mathbf{x})$ of the pixel space are shown here. The beginning and ending images of each row locate in different manifolds. The intermediate results are sampled by crossing between disconnected manifolds. The discontinuity causes weird generation in the transition, marked with yellow boxes (**first row**: the cat misses an ear, and its head is crooked; **second row**: white spots on the nose; **third row**: two bodies share one head).

In this paper, we utilize the adversarial sample search method, the iterative fast gradient sign method (I-FGSM) (Kurakin et al., 2016) to conduct latent transform and mine optimization-friendly $\mathbf{z}$ samples in the GAN training process, which matches up with the procedure of adversarial training. While most existing works make perturbations on pixel space (real samples or generated samples), instead we focus on mining the latent space and making perturbations on latent samples.

## 3 PROPOSED METHOD

This section presents AdvLatGAN-z to achieve a transformation $z^*(\cdot)$, and further AdvLatGAN-qua and AdvLatGAN-div to obtain generation mapping $G$ with better generation quality (i.e., -qua) and diversity (i.e., -div). Both methods are derived from the iterative updating in latent space.

### 3.1 PRELIMINARIES

We introduce adversarial samples and adversarial training used to derive $z^*(\mathbf{z})$ and devise training algorithms, and the regularization technique in MSGAN (Mao et al., 2019) for mode seeking.

**Adversarial samples and adversarial training.** Adversarial training aims to defend against adversarial attacks. It uses adversarial samples to calculate the optimized loss. In classification, for each image-label pair $(\mathbf{x}_i, y_i) \in S$ from the labeled training set, the adversarial samples $\mathbf{x}'$ is defined as:

$$\mathbf{x}' = \underset{||\mathbf{x}'-\mathbf{x_i}||_p \leq \epsilon}{\arg\max} \ l(f_\theta(\mathbf{x}'), y_i) \tag{1}$$

Here $\epsilon$ is the radius of the closed ball to constrain perturbation. While $f_\theta$ and $l$ denote the attack target (a neural net) and the corresponding loss (e.g. cross-entropy loss). $p$ denotes the norm dimension. Given in total $n$ adversarial samples, adversarial training is fulfilled by the optimization:

$$\theta = \min_\theta \frac{1}{n} \sum_{i=1}^{n} l(f_\theta(\mathbf{x}'_i), y_i) \tag{2}$$

**Mode coverage by regularizing distance between generated samples.** MSGAN (mode seeking GAN) (Mao et al., 2019) mitigates mode collapse in conditional generation setting by introducing a regularization term for training $G$ over two latent samples $\mathbf{z}_1$ and $\mathbf{z}_2$ as follows. The hope is that the two generated targets shall be far away from each other as much as possible:

$$\mathcal{L}_{ms} = \frac{d_I\left(G(c, \mathbf{z}_1), G(c, \mathbf{z}_2)\right)}{d_z\left(\mathbf{z}_1, \mathbf{z}_2\right)} \tag{3}$$

where $c$ is the condition vector for generation, and $d_I$ and $d_z$ are the distance metrics in the target (image) space and latent space, respectively. It encourages the distance to be maintained or even enlarged in the target space to avoid the generated samples being trapped into an aggregation.

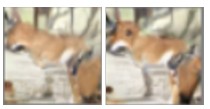 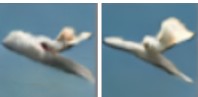 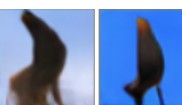 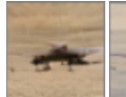 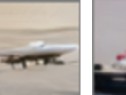 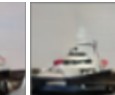

Figure 4: Results of latent sample transform on the SNGAN model pre-trained on STL-10. The original generated images are shown on the left, and the images generated with the newly found $\mathbf{z}$s after 100 steps iteration of I-FGSM are shown on the right. We notice that the right figures contain richer colors and more realistic details, with the main content remaining consistent. We measure generation quality in a batch by Inception Score, which increases from 6.944 to 7.475.

By integrating the regularization term with the original objective function of cGANs (Mirza & Osindero, 2014), the final objective for minimization in MSGAN for training $G$ and $D$ is given by:

$$\begin{aligned} \mathcal{L} &= \mathcal{L}_{ori} + \lambda_{ms}\mathcal{L}_{ms} \\ \mathcal{L}_{ori} &= \mathbb{E}_{c,\mathbf{x}}[\log D(c,\mathbf{x})] + \mathbb{E}_{c,\mathbf{z}}[\log(1 - D(c,G(c,\mathbf{z})))] \end{aligned} \tag{4}$$

where $\mathcal{L}_{ori}$ denotes the original objective, $\mathbf{x}$ denotes real data, and $\lambda_{ms}$ is the weight hyper-parameter. Note since Eq. 3 is aimed to be maximized hence one can interpret $\lambda_{ms}$ to be negative.

### 3.2 LATENT SPACE ADVERSARIAL SAMPLE SEARCH FOR GENERATION QUALITY

We show how to implicitly transform the latent raw samples from Gaussian (or other popular forms e.g. uniform) to a new distribution via the idea of searching adversarial samples as introduced above. This transform breaks the dilemma in diversity and quality where the bottleneck is due to the continuous latent distribution and a continuous mapping function. Moreover, conducting the implicit transform in training to obtain the new (probably disconnected) latent distribution for each iteration can compensate for the difficulty for generators to align the pace of generators and discriminators[1], which cannot map a continuous distribution to a disconnected one with multiple modes.

Specifically, we design the following constrained optimization task that encourages the new $\mathbf{z}^*$ to be close enough[2] to $\mathbf{z}$. Its prediction score (as real data) by the discriminator $D$ should be as high as possible to improve the generation quality. Recall otherwise the given $\mathbf{z}_0$ may correspond to a poor generation as discussed in Sec. 1. Note here that both $G$ and $D$ are fixed.

$$\mathbf{z}^*(\mathbf{z}_0) = \underset{\mathbf{z} \in \{\mathbf{z} | d(\mathbf{z}_0, \mathbf{z}) \le \epsilon\}}{\arg\min} \log(1 - D(G(\mathbf{z}))) \tag{5}$$

where $\mathbf{z}_0$ denotes the original latent vectors, $\mathbf{z}^*(\mathbf{z}_0)$ denotes the newly computed $\mathbf{z}$, $d(\cdot)$ denotes the distance of two vectors which is an $\ell_\infty$ distance in this paper, in line with the popular protocol in adversarial attack literature (Goodfellow et al., 2014b).

By borrowing the well-developed tools in adversarial attack, the solution of the above problem can be estimated by the so-called iterative fast gradient sign method (I-FGSM) (Kurakin et al., 2016):

$$\mathbf{z}_{i+1} \leftarrow \mathbf{z}_i - \epsilon sgn\{\nabla_{\mathbf{z}_i}\log(1 - D(G(\mathbf{z}_i)))\} \tag{6}$$

From the distribution perspective, the transform can be characterized by:

$$p_z^* = \arg \min_{D(p_z, p_z^0) \le \epsilon} E_{\mathbf{z} \sim p_z}[\log(1 - D(G(\mathbf{z})))] \tag{7}$$

where $p_z^0$ denotes the original latent distribution, $p_z^*$ denotes the newly searched one. $D(p_z, p_z^0) \le \epsilon$ means that there is a limit to the distribution variation. A newly searched $\mathbf{z}$ can be considered as a sample from $p_z^*$ if the sampling is within the perturbation circle. Fig. 4 shows the results by the pre-trained SNGAN (Miyato et al., 2018) (i.e. the given $G$ and $D$) on STL-10 (Coates et al., 2011).

We develop a novel GAN training algorithm by conducting the above latent distribution shift to the initial sampling of Gaussian in $D$ updating iterations, while the rest of the training algorithm remains consistent with vanilla GAN (Goodfellow et al., 2014a). We present the algorithm in Alg. 1 which we call Adversarial Latent GAN (AdvLatGAN-qua) with a post meaning quality. The latent tranform is first computed using the I-FGSM to find the adversarial samples in latent space, then $D$ and $G$ are updated. The iteration continues until enough steps or convergence of $G$.

---

[1]In literature, it is widely recognized that it is much more challenging to train the generator than discriminator.
[2]We do not want to departure too much from the basic GAN training protocol, and also the transform shall keep mode consistent to maintain overall diversity.

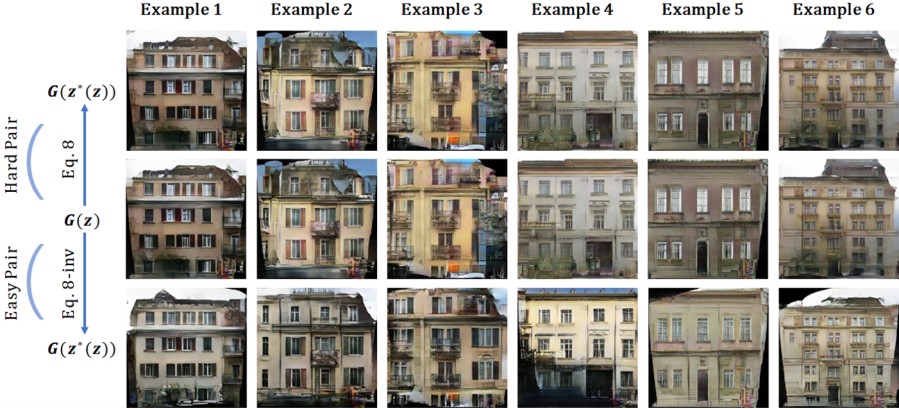

Figure 5: Each of the six columns shows two compared pairs of generated example (first pair: 1st + 2nd row; second pair: 3rd row + 2nd row) by our diversity driven iterative transform scheme in latent space with the same number of iterations to obtain $z^*$ from initial $z$. **Middle row:** generation by vanilla latent sampling $z$; **Top:** by latent sample transformed by Eq. 8; **Bottom:** by latent sample transformed by Eq. 8's inverse form. It shows using Eq. 8 generates more similar image pairs serving as hard samples for training, rather than using its inverse form (see quantitative results in Table 1).

The above latent transform model does not consider how to avoid mode collapse, as each time only one latent sample is considered for optimization in isolation. Similar constrained optimization formulation can be designed to improve the generation diversity, as shown in the following subsection.

## 3.3 LATENT SPACE ADVERSARIAL SAMPLE SEARCH FOR GENERATION DIVERSITY

We take a hard sample mining perspective to improve mode coverage. Given a sample $\mathbf{z}_0$ from the raw latent distribution, we aim to search its paired sample $\mathbf{z}^*$ to form a hard sample pair, in the sense that their generations are close in the target space given $G$. We consider two aspects. First, as hard samples, the paired samples shall still belong to the same category $\mathbf{c}$ which can be modeled with conditional GAN. Second, their distance in the latent space shall also be not too close otherwise, they may be aggregated in target space due to continuous mapping. In other words, hard samples causing mode collapse shall be those close-in target space while apart in the latent space, when $G$ is given. Based on the above considerations, we adopt the form of Eq. 3 to design our search procedure:

$$\mathbf{z}^*(\mathbf{z}_0) = \underset{\mathbf{z} \in \{\mathbf{z} | d(\mathbf{z}_0, \mathbf{z}) \leq \epsilon\}}{\arg \min} \frac{d_I(G(\mathbf{c}, \mathbf{z}_0), G(\mathbf{c}, \mathbf{z}))}{d_z(\mathbf{z}_0, \mathbf{z})} \tag{8}$$

The symbols and terms are similar to Eq. 3. Akin to AdvLatGAN-qua, we use I-FGSM to obtain $\mathbf{z}^*$ and again use $\ell_\infty$ to control change magnitude. For each update step, we randomly select the first latent vector $\mathbf{z}_0$, and the second vector $\mathbf{z}^*$ is searched by $n$ iterations of the I-FGSM method under the $\ell_\infty$ restriction starting from the neighborhood of $\mathbf{z}_0$, and the iterations are guided by Eq. 8.

Table 1: Ratio between image and latent distance by Eq. 8 to derive new $\mathbf{z}^*$. *-inv* means using inverse form.

| Dataset | Eq. 8 | Eq. 8-inv |
|---------|-------|-----------|
| CIFAR-10 | 0.241 | 4.023 |
| FACADES | 0.3785 | 1.3385 |

Table 1 shows the effectiveness of the above paired complex sample seeking approach, with experiments on CIFAR-10 (Krizhevsky et al., 2009) and Facades (Tyleček & Šára, 2013) with DC-GAN (Radford et al., 2016) as backbone. The visual results on Facades are shown in Fig. 5. We can see that the pair obtained by solving Eq. 8 tend to collapse (as they are hard samples), while the opposite leads to better diversity by using the inverse form of Eq. 8.

The above experiments show the effectiveness of our iteration method, and we further introduce it into the initial training process of MSGAN (Mao et al., 2019). In MSGAN, the regularization term 3 is used to regularize the model optimization process, where the selection of the pair is entirely random, without using any heuristic information. We instead use the $\mathbf{z}$ pair that are more inclined to collapse by randomly taking one $\mathbf{z}$ and iterating by the method mentioned above to get another $\mathbf{z}$. We keep the other settings of the algorithm unchanged.

Table 3: Post-training latent sampling improvement with two architectures on STL-10.

| Architecture | Method | SNGAN (two initials) | | | | WGANGP (two initials) | | | |
|---|---|---|---|---|---|---|---|---|---|
| | | IS($\uparrow$) | FID($\downarrow$) | IS($\uparrow$) | FID($\downarrow$) | IS($\uparrow$) | FID($\downarrow$) | IS($\uparrow$) | FID($\downarrow$) |
| ResNet | Original | 7.244 | 64.573 | 6.285 | 46.076 | 7.287 | 73.625 | 6.791 | 47.765 |
| | AdvLatGAN-z | **8.116** | **62.184** | **6.833** | **43.099** | **7.743** | **73.443** | **7.369** | **45.121** |
| DCGAN | Original | 6.183 | 46.701 | 7.344 | 64.289 | 6.701 | 48.454 | 7.112 | 69.263 |
| | AdvLatGAN-z | **6.748** | **43.386** | **7.860** | **64.165** | **7.376** | **46.490** | **7.767** | **68.929** |

By using Eq. 8 to generate adversarial sample pairs to address mode collapse, we finally develop our new diversity enhancing GAN training approach AdvLatGAN-div as shown in Alg. 2 in the appendix, which are further enhanced by post-training as discussed in experiments.

## 4 EXPERIMENTS

Experiments are performed on a single GPU of GeForce RTX 3090. All datasets used in experiments are publicly available.

### 4.1 EXPERIMENTAL SETUP

We validate the effectiveness of the proposed methods for two parts, respectively for effective latent sampling improvement $z^*$ and better generation mapping $G$. In particular, five methods can be derived for different targets, among which AdvLatGAN-qua+ and AdvLatGAN-div+ are the final full version of our contributions. We present the methods below:

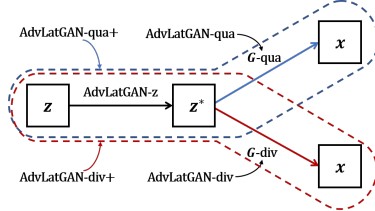

Figure 6: The logic of our 5 variant methods in Sec. 4.1. Blue and red is for quality and diversity.

Table 2: Components of our 5 variant methods in Sec. 4.1.

| | Latent Transform $z^*$ | Improved mapping $G$ |
|---|---|---|
| i) | ✓ | ✗ |
| ii) | ✗ | ✓ |
| iii) | ✗ | ✓ |
| iv) | ✓ | ✓ |
| v | ✓ | ✓ |

**i) AdvLatGAN-z:** post-training latent sampling improvement fighting against quality discontinuity; **ii) AdvLatGAN-qua:** GAN training algorithm for better generative quality using in-training latent sampling improvement; **iii) AdvLatGAN-div:** GAN training algorithm for a more diverse generation using in-training latent sampling improvement; **iv) AdvLatGAN-qua+:** both in and post-training latent sampling improvement for generation quality; **v) AdvLatGAN-div+:** both in and post-training latent sampling improvement for generation diversity.

For illustrative comparison, we show the schematic overview of these methods in Fig. 6 and Table 2.

We adopt Inception Score (Salimans et al., 2017) and Fréchet Inception Distance (Heusel et al., 2017) to evaluate the performance, which utilizes the classification results of the generated images by the Inception Network to evaluate the performance of the model. Following (Richardson & Weiss, 2018), JSD is adopted to measure the similarity between generated images and real images, which first clusters real images and generated images into the same number of classes using the K-means algorithm, then calculate the similarity between real and fake clusters w.r.t. the clustering property. We also adopt recently proposed evaluation metrics density/coverage (Naeem et al., 2020), which separates the fidelity evaluation and the diversity evaluation.

### 4.2 ADVLATGAN-Z: POST-TRAINING LATENT SAMPLING IMPROVEMENT

This section shows the effectiveness of the post-training latent sampling improvement AdvLatGAN-z (denoted as $z^*$ in previous sections, which is an implicit transform on the original latent sampling from Gaussian), which aims to mitigate the issue of quality discontinuity.

**Results on Synthetic Data.** In line with (Metz et al., 2017), we apply the proposed iterative strategy guided by Eq. 5 in both Grid and Ring datasets to evaluate its performance. By updating latent space variables, we achieve high-quality generated samples. The results in Fig. 7 show the effectiveness of our method.

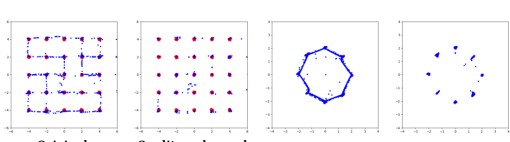

Figure 7: Post-training latent sampling improvement on Grid and Ring by AdvLatGAN-qua.

**Results on MNIST.** MNIST contains 70,000 images of handwritten digits (LeCun et al., 1998), which can be generated from a two-dimensional latent space in which the iterative latent distribution

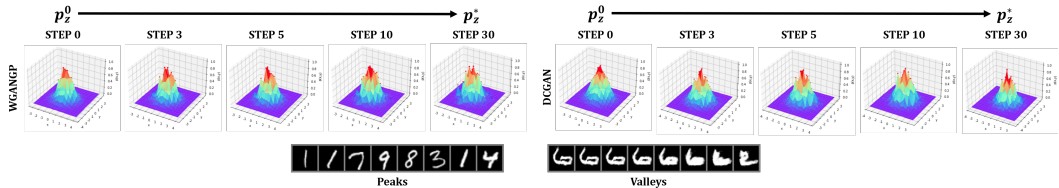

Figure 8: Latent distribution transform with WGAN-GP/DCGAN as backbone on MNIST. As the I-FGSM iteration continues, the latent distribution deviates from the standard Gaussian, and contains more valleys and peaks which can be mapped into generated images as shown in the bottom. Note the generations by the valley points are of low quality and they can be effectively avoided by our scheme as the sampling probability is low (valley) in our transformed distribution.

.

Table 4: The results of AdvLatGAN-qua and AdvLatGAN-qua+ for CIFAR10.

| Architecture | Inception Score($\uparrow$) | | | Fréchet Inception Distance($\downarrow$) | | |
|---|---|---|---|---|---|---|
| | vanilla | AdvLatGAN-qua | AdvLatGAN-qua+ | vanilla | AdvLatGAN-qua | AdvLatGAN-qua+ |
| DCGAN | $5.92 \pm 0.05$ | $6.21 \pm 0.07$ | $\mathbf{6.58 \pm 0.34}$ | $46.4 \pm 0.7$ | $41.7 \pm 1.0$ | $\mathbf{40.1 \pm 1.5}$ |
| WGAN | $6.63 \pm 0.05$ | $7.21 \pm 0.04$ | $\mathbf{7.76 \pm 0.07}$ | $32.8 \pm 0.5$ | $27.3 \pm 0.7$ | $\mathbf{27.2 \pm 0.5}$ |
| WGAN-GP | $7.47 \pm 0.09$ | $7.60 \pm 0.06$ | $\mathbf{8.59 \pm 0.10}$ | $24.7 \pm 0.2$ | $22.6 \pm 0.4$ | $\mathbf{18.3 \pm 1.1}$ |
| SNGAN | $7.29 \pm 0.09$ | $7.58 \pm 0.03$ | $\mathbf{8.13 \pm 0.06}$ | $25.5 \pm 0.3$ | $22.3 \pm 0.5$ | $\mathbf{21.9 \pm 0.3}$ |
| LSGAN | $5.87 \pm 0.14$ | $6.13 \pm 0.27$ | $\mathbf{6.38 \pm 0.18}$ | $49.3 \pm 2.2$ | $42.8 \pm 1.3$ | $\mathbf{41.0 \pm 1.3}$ |
| WGAN-div | $7.43 \pm 0.02$ | $0.02 \pm 0.03$ | $\mathbf{8.67 \pm 0.11}$ | $23.8 \pm 0.3$ | $20.6 \pm 0.4$ | $\mathbf{16.0 \pm 0.5}$ |
| ACGAN | $6.02 \pm 0.28$ | $6.06 \pm 0.21$ | $\mathbf{6.21 \pm 0.30}$ | $59.5 \pm 1.5$ | $53.7 \pm 1.6$ | $\mathbf{53.1 \pm 0.4}$ |

Table 5: The results of AdvLatGAN-qua and AdvLatGAN-qua+ for STL10

| Architecture | Inception Score($\uparrow$) | | | Fréchet Inception Distance($\downarrow$) | | |
|---|---|---|---|---|---|---|
| | vanilla | AdvLatGAN-qua | AdvLatGAN-qua+ | vanilla | AdvLatGAN-qua | AdvLatGAN-qua+ |
| DCGAN | $7.18 \pm 0.09$ | $7.33 \pm 0.07$ | $\mathbf{7.79 \pm 0.06}$ | $61.2 \pm 1.2$ | $56.3 \pm 1.0$ | $\mathbf{54.6 \pm 1.3}$ |
| WGAN | $6.51 \pm 0.07$ | $7.62 \pm 0.04$ | $\mathbf{8.16 \pm 0.27}$ | $73.0 \pm 0.2$ | $51.0 \pm 0.6$ | $\mathbf{49.1 \pm 1.2}$ |
| WGAN-GP | $8.86 \pm 0.05$ | $8.90 \pm 0.05$ | $\mathbf{10.32 \pm 0.34}$ | $37.4 \pm 0.4$ | $34.2 \pm 0.9$ | $\mathbf{26.7 \pm 1.1}$ |
| SNGAN | $8.49 \pm 0.09$ | $8.63 \pm 0.08$ | $\mathbf{9.37 \pm 0.05}$ | $36.8 \pm 0.4$ | $34.5 \pm 0.21$ | $\mathbf{30.9 \pm 0.4}$ |
| LSGAN | $7.08 \pm 0.12$ | $7.16 \pm 0.15$ | $\mathbf{7.55 \pm 0.18}$ | $62.9 \pm 2.2$ | $58.5 \pm 1.3$ | $\mathbf{57.1 \pm 1.5}$ |
| WGAN-div | $8.82 \pm 0.02$ | $9.00 \pm 0.01$ | $\mathbf{10.68 \pm 0.17}$ | $37.7 \pm 0.2$ | $32.0 \pm 0.6$ | $\mathbf{24.3 \pm 1.0}$ |

change is illustrated in Fig. 8. We respectively conduct the experiments for the trained DCGAN model and WGAN-GP (Gulrajani et al., 2017) model, with single-step iteration constraint using $\ell_\infty$ and iterated step size $\epsilon$ is set to 0.03. To better reflect changes in the distribution, we only show the first 30 iterations that can reflect larger changes. For WGAN test, Fig. 8 shows the transformed latent distribution which contains more peaks and valleys and is different from the raw Gaussian.

**Results on CIFAR-10 and STL-10.** We run experiments on CIFAR-10 (Krizhevsky et al., 2009) and STL-10 (Coates et al., 2011). CIFAR-10 has been widely used with more diverse and closer to the quality of high-resolution images than MNIST. STL-10 has a higher resolution and richer image representation. We use implicit transform in the latent space for pre-trained DCGAN and SNGAN respectively, with the single-step iteration constraint using $\ell_\infty$ and iteration step size $\epsilon$ is set to 0.01. We conduct 100 steps for each iteration process. Inception Score and Fréchet Inception Distance are used to evaluate the generation quality, which are calculated over 1500 generated images for STL-10 and over 3000 images for CIFAR-10. For each of the four cases, we use two different initializations, each initialized randomly and not explicitly chosen. As shown in Table 3, our method has a significant impact on generation quality, with a maximum improvement of 12.03% by Inception Score.

**Results on AFHQ and FFHQ.** To show the effectiveness of AdvLatGAN-z to solve issues in Fig. 3, we evaluate on AFHQ (Choi et al., 2020) and FFHQ (Karras et al., 2019b) based on the StyleGAN2-ada (Karras et al., 2020) model. Note that AdvLatGAN-z works at the sample level (or distribution level as a distribution shift characterized by sample transforms) in the latent space. In the Fig. 3 setting, when the sampling encounters invalid examples, AdvLatGAN-z is able to transform the bad $z$ to $z^*(z)$ with better ability to avoid bad generation. Here we verify whether invalid samples can be improved with AdvLatGAN-z. We select six bad generations with the first three correspond to those in Fig. 3 and conducting AdvLatGAN-z to their latent vectors. The results show that AdvLatGAN-z can effectively fix or avoid image defects, as shown in Fig. 9.

### 4.3 ADVLATGAN-QUA/DIV: IMPROVING THE GENERATION MAPPING

This section shows the effectiveness of the GAN training algorithms, which aims to improve the generation mapping $G$ for better performance respectively on quality and diversity.

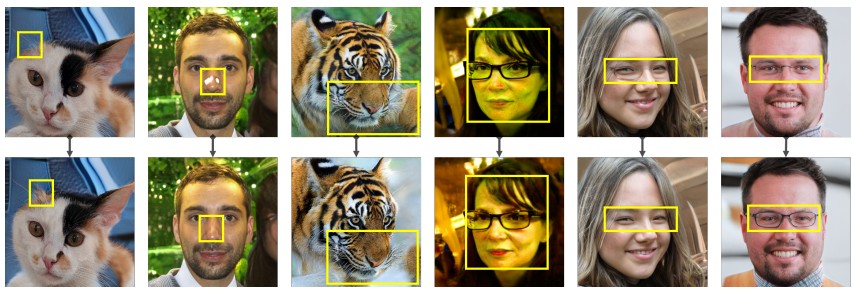

Figure 9: Results of AdvlatGAN-z on AFHQ and FFHQ. The first row are bad generations (the first three correspond to those in Fig. 3) with the defects as follows: **first column**: the cat misses an ear; **second column**: white spots on the nose; **third column**: two bodies share one head; **fourth column**: green face; **fifth and sixth columns**: semi-existing glasses. The second row are results under AdvLatGAN-z. The results show that AdvLatGAN-z can effectively mitigate image defects.

.

Table 6: FID and JSD results on CIFAR-10.

| Metrics | Models | overall | airplane | automobile | bird | cat | deer | dog | frog | horse | ship | truck |
|---|---|---|---|---|---|---|---|---|---|---|---|---|
| FID(↓) | MSGAN | 30.225 | 73.083 | **69.518** | 78.258 | 74.525 | 57.778 | 86.831 | 63.287 | 69.705 | 69.994 | 66.434 |
|  | AdvLatGAN-div+ | **26.763** | **70.142** | 69.642 | **75.922** | **66.507** | **53.725** | **83.912** | **54.893** | **64.943** | **69.715** | **62.173** |
| JSD(↓) | MSGAN | 0.00783 | 0.03004 | 0.02737 | 0.02861 | **0.02571** | 0.02722 | **0.02670** | 0.03381 | 0.04340 | 0.03796 | 0.04093 |
|  | AdvLatGAN-div+ | **0.00542** | **0.02694** | **0.02201** | **0.02795** | 0.02623 | **0.02835** | 0.02805 | **0.02776** | **0.03548** | **0.03635** | **0.03210** |
| density(↑) | MSGAN | 0.350 | **0.338** | 0.216 | **0.308** | 0.635 | **0.532** | 0.272 | 0.665 | 0.296 | 0.331 | 0.224 |
|  | AdvLatGAN-div+ | **0.516** | 0.333 | **0.255** | 0.267 | **0.679** | 0.511 | **0.318** | **0.674** | **0.300** | **0.353** | **0.231** |
| coverage(↑) | MSGAN | 0.568 | 0.651 | 0.690 | 0.460 | 0.674 | 0.736 | 0.452 | 0.776 | **0.779** | 0.817 | **0.612** |
|  | AdvLatGAN-div+ | **0.726** | **0.677** | **0.752** | **0.464** | **0.748** | **0.797** | **0.466** | **0.843** | 0.743 | **0.877** | 0.604 |

Table 7: FID and JSD results on STL-10.

| Metrics | Models | overall | airplane | bird | car | cat | deer | dog | horse | monkey | ship | truck |
|---|---|---|---|---|---|---|---|---|---|---|---|---|
| FID(↓) | MSGAN | 67.849 | **92.021** | 125.723 | **108.434** | 118.938 | 111.784 | 133.680 | 140.486 | 121.907 | 101.232 | 101.059 |
|  | AdvLatGAN-div+ | **65.349** | 92.169 | **124.155** | 109.3590 | **118.385** | **105.966** | **132.681** | **139.372** | **117.149** | **95.298** | **99.590** |
| JSD(↓) | MSGAN | 0.00661 | 0.02968 | 0.02818 | **0.03177** | 0.03849 | 0.03522 | 0.03107 | 0.03996 | 0.03498 | 0.02681 | 0.03157 |
|  | AdvLatGAN-div+ | **0.00423** | **0.02852** | **0.02717** | 0.03638 | **0.03518** | **0.03773** | **0.03132** | **0.03198** | **0.03891** | **0.02239** | **0.02751** |
| density(↑) | MSGAN | 0.332 | 0.192 | 0.264 | 0.089 | 0.598 | 0.399 | 0.397 | 0.150 | 0.251 | 0.251 | 0.116 |
|  | AdvLatGAN-div+ | **0.443** | **0.235** | **0.328** | **0.111** | **0.733** | **0.456** | **0.429** | **0.194** | **0.353** | **0.165** | **0.126** |
| coverage(↑) | MSGAN | 0.361 | 0.383 | **0.364** | 0.169 | 0.418 | 0.306 | 0.289 | 0.271 | 0.280 | 0.329 | 0.226 |
|  | AdvLatGAN-div+ | **0.396** | **0.408** | 0.329 | **0.226** | **0.508** | **0.424** | **0.309** | **0.390** | **0.335** | **0.395** | **0.395** |

**AdvLatGAN-qua for Quality Improvement.** We conduct experiments on CIFAR-10 and STL-10, using the mainstream architectures including DCGAN, WGAN, WGAN-GP, SNGAN, LSGAN, WGAN-div (Wu et al., 2018) and ACGAN (Kang et al., 2021). We do not include ACGAN in the STL-10 setting because it does not work on unlabeled dataset. We adversarially train GAN using a quality-enhanced strategy previously mentioned as AdvLatGAN-qua, and the results show that the proposed method can greatly improve the quality of generated images. IS and FID are adopted to evaluate the performance of the model. We conduct only one latent iteration to save computation cost per generator step, and more details will be given in the appendix. We also try to use implicit transform after training to improve the generation quality further. The result is shown in Table 4 and Table 5. It can be seen that the proposed method achieves a significant improvement in generative performance compared to that without the latent space transformation strategy. In the WGAN STL-10 setting, IS is improved from 6.51 to 8.16, and FID from 73.0 to 49.1.

**AdvLatGAN-div for Diversity Improvement.** We conduct experiments on both CIFAR-10 and STL-10 for conditional generation and adopt FID, JSD, density and coverage as the metrics, respectively for the entire set of generated images and each class. We test the training results on the basis of implicit latent transform to fight against quality discontinuity. We can find that the proposed method is prior over MSGAN reflected by JSD and FID metrics. The results are shown in Table 6 and Table 7. For the overall set which models are trained, AdvLatGAN-div+ has achieved at most 11.5% improvement in FID, 36.0% in JSD, 47.4% in density and 27.8% in coverage.

## 5 CONCLUSION

This work begins with two basic observations in GANs from the perspective of continuous mapping. Based on the observations, we introduce the adversarial attack techniques in GAN to explore latent space and derive novel training algorithms to improve the mapping property. With the efforts of both latent transform and mapping improvement, the proposed AdvLatGAN has shown promising ability for achieving generation with both quality and diversity verified by extensive experiments.

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

APPENDIX

## A   DETAILED ALGORITHM FOR ADVLATGAN-QUA AND ADVLATGAN-DIV

The specific algorithm for AdvLatGAN-qua in Sec. 3.2 is presented in Alg. 1.

---

**Algorithm 1 AdvLatGAN-qua** $t_z, t_D$: the number of steps, other symbols are consistent with vanilla GAN.

---

**Input:** $p_z$ e.g. Gaussian, $p_r$ e.g. real images distribution; randomly initialized $G$ and $D$;
**Output:** trained generator $G$ and discriminator $D$;
1: **while** $G$ has not converged **do**
2:     *// update D:*
3:     **for** $i = 1$ to $t_D$ **do**
4:         Sample $\{\mathbf{z}_0^{(k)}\}_{k=1}^m \sim p_z$;
5:         *// the proposed sampling shift (t-step I-FGSM):*
6:         **for** $i = 1$ to $t_z$ **do**
7:             Obtain $\{\mathbf{z}_i^{(k)}\}_{k=1}^m$ by Eq. 5;
8:         Sample $\{\mathbf{x}_i^{(k)}\}_{k=1}^m \sim p_r$;
9:         Calculate $loss_D$ with $\{\mathbf{x}_i^{(k)}\}_{k=1}^m$ and $\{\mathbf{z}_t^{(k)}\}_{k=1}^m$;
10:         Update $D$ with $loss_D$;
11:     *// update G:*
12:     Sample $\{\mathbf{z}^{(k)}\}_{k=1}^m \sim p_z$;
13:     Calculate $loss_G$ with $\{\mathbf{z}^{(k)}\}_{k=1}^m$;
14:     Update $G$ with $loss_G$;

---

The specific algorithm for AdvLatGAN-div in Sec. 3.3 is presented in Alg. 2.

---

**Algorithm 2 AdvLatGAN-div** $t_z, t_D$: the number of steps, other symbols are consistent with vanilla GAN.

---

**Input:** $p_z$ e.g. Gaussian, $p_r$ e.g. real images distribution; $\lambda_{ms}$ is a hyper-parameter; randomly initialized $G$ and $D$;
**Output:** trained generator $G$ and discriminator $D$;
1: **while** $G$ has not converged **do**
2:     *// update D:*
3:     **for** $i = 1$ to $t_D$ **do**
4:         Sample $\{\mathbf{z}_i^{(k)}\}_{k=1}^m \sim p_z$;
5:         Sample $\{(\mathbf{x}_i^{(k)}, c_i^{(k)})\}_{k=1}^m \sim p_r$;
6:         Compute $loss_D$ with $\{(\mathbf{x}_i^{(k)}, c_i^{(k)}), \mathbf{z}_i^{(k)}\}_{k=1}^m$;
7:         Update $D$ with $loss_D$;
8:     *// update G:*
9:     Randomly generate labels $\{c^{(k)}\}_{k=1}^m$;
10:     Sample $\{\mathbf{z}_0^{(k)}\}_{k=1}^m \sim p_z$;
11:     *// the proposed sampling shift (t-step I-FGSM):*
12:     **for** $i = 1$ to $t_z$ **do**
13:         Obtain $\{\mathbf{z}_i^{(k)}\}_{k=1}^m$ by Eq. 8;
14:     Calculate $loss_G$ and $loss_{ms}$ (Eq. 3) with $\{c^{(k)}\}_{k=1}^m$, $\{\mathbf{z}_0^{(k)}\}_{k=1}^m$ and $\{\mathbf{z}_t^{(k)}\}_{k=1}^m$;
15:     $loss \leftarrow loss_G + \lambda_{ms} loss_{ms}$;
16:     Update $G$ with $loss$;

---

## B   RELATIONSHIP WITH ADVERSARIAL ATTACKS AND DEFENSES

Our approach is closely related to adversarial attacks and defenses. As for works in adversarial attacks, the fundamental standpoint is that deep neural networks' performance can vary significantly in the face of small perturbations. This scenario can be very similar to our starting point as described

in Sec 1 that a small perturbation of the latent vectors can lead to massive quality variation. Thus it is quite natural to combine the adversarial attacks and defenses with generative adversarial networks, and specifically, we utilize the latent transform using I-FGSM iterative method to overcome the quality discontinuity.

Our GAN training algorithms AdvLatGAN-qua and AdvLatGAN-div can be analogous to adversarial training techniques in the adversarial defense field. Adversarial training is a proactive defense approach that strengthens the model against attacks or enhances its performance by modifying the inputs to use adversarial samples to train the model, making the model naturally defensive against attacks. This process involves conducting adversarial attacks during training and can be explained as mining targeted optimization-friendly samples for training. While in AdvLatGAN-qua and AdvLatGAN-div, we conduct $\mathbf{z}$ iterations in the GAN training referencing attack method, which can be viewed as the same process as adversarial training. AdvLatGAN introduces the latent vector $\mathbf{z}$ iterations into the bi-level optimization between the generator $G$ and the discriminator $D$. The training involves iterations of three components $\mathbf{z}$, $G$, and $D$, among which the introduced $\mathbf{z}$ iterations can mine the latent space during training, to some extent leading to the superiority of the method.

## C  DESCRIPTION OF CONSTRAINTS FOR ITERATION

There are two basic types of constraint used in the field of adversarial attack: 1) $\ell_2$ norm constraint; 2) $\ell_\infty$ norm constraint. In this paper, we follow the well-known attack method, iterative fast gradient sign method (I-FGSM) (Kurakin et al., 2016) to conduct the iterations of latent variables and we adopt the $\ell_\infty$ norm as our constraint.

## D  DIFFERENCES AND COMPARISONS TO OTHER ALGORITHMS

We present the differences between our approach and other previous works, involving two topics i.e. latent exploration and GAN model with adversarial training. An overview is presented in Table 8. Meanwhile, we conduct experiments to show our superiority over other representative works.

### D.1  LATENT EXPLORATION

**EvolGAN (Roziere et al., 2020), Tarsier (Roziere et al., 2021).** These two models both use the well-known quality evaluator Koncept512 (which is actually a classifier) to guide the latent iterations. Their drawback is that they do not exploit the information of the real distribution, which leads to that the optimization criterion of sampling quality is not for the proximity to the real distribution, but to match the data trained for the Koncept512 classifier. Our approach uses a discriminator to guide the latent variable updating, which makes full use of the information from the real distribution, while further achieving novel training algorithm to improve the generative mapping.

**DDLS (Che et al., 2020).** DDLS analyzes the sample-improving approach from the perspective of the energy-based model, which differs from ours in terms of methodology. Besides, it didn't notice that when we enhance the quality, we hope that the image is changed as little as possible. Combing the above two points, we explain the problem from the perspective of attack, which is quite different from the specialized sampling approach Langevin Dynamics used in (Che et al., 2020).

### D.2  GAN MODEL WITH ADVERSARIAL TRAINING

Our GAN training algorithm can be considered as a kind of adversarial training from the perspective of adversarial attack and defense, and we differ from several previous works that combine adversarial training and GAN.

**ASGAN (Liu et al., 2021), FastGAN (Zhong et al., 2020), Rob-GAN (Liu & Hsieh, 2019).** These methods add perturbation to the real images (the latter is an improvement in the loss function compared to the former), while we focus on the latent space, perturbing latent space vectors.

**Robust GAN training (Zhou & Krähenbühl, 2018).** This algorithm shows that a robust discriminator can benefit training and this robustness only need to be enforced in expectation over the generated samples, again without focusing on the latent space compared to us.

Table 8: Differences from Other Algorithms

| Viewpoint | Method | Different point | Detailed Description | |
|---|---|---|---|---|
| | | | Theirs | Ours |
| Latent Exploration | EvolGAN | Guide of the transform | Quality estimator Koncept512 | $D \cdot G$ |
| | | Achieving method | Evolutionary Algorithm | I-FGSM iterations |
| | Tarsier | Guide of the transform | Quality estimator Koncept512 | $D \cdot G$ |
| | | Achieving method | Diagonal Covariance Matrix Adaptation | I-FGSM iterations |
| | | Targeted Task | Super-resolution generation tasks | General generation tasks |
| | AE-OT-GAN | Timing for the transform | Before training G and D | After training G and D |
| | | Achieving method | First train an Auto-Encoder to learn a latent distribution then use optimal trasport to achieve the transform | I-FGSM iterations |
| | | Calculating cost | Need to train an Auto-Encoder for finding the latent distribution | No extra networks are trained (guided by G and D) |
| | DDLS | Basis of theoretical analysis | Energy-based theory | General GAN theory |
| | | Achieving method | Markov Chain Monte Carlo | I-FGSM iterations |
| AdvLatGAN-qua | vanilla GAN | $z$ in the training of D | Samples from Gaussian | Transformation of the original sampling |
| AdvLatGAN-div | MSGAN | $z$ pair used in ms-regularization term | Individually sampled from Gaussian | Randomly choose $z_1$, then transform $z_1$ to get $z_2$, forming a pair |
| Adversarial Training | Common differences | Objects to which perturbations are added | $x$ | $z$ |

Table 9: Experimental comparisons over GAN with adversarial training evaluated by **Inception Score.**

| Framework | Evolgan (a=0,b=80) | Evolgan (a=0.5,b=80) | Evolgan (a=0.5,b=20) | Evolgan (a=1,b=20) | DDLS | AdvLatGAN-z |
|---|---|---|---|---|---|---|
| DCGAN | 5.815 | 5.887 | 5.825 | 5.791 | 5.502 | **6.086** |
| WGAN | 6.528 | 6.496 | 6.521 | 6.546 | 2.607 | **7.334** |
| WGAN-GP | 7.349 | 7.316 | 7.441 | 7.236 | 7.074 | **8.279** |
| SNGAN | 7.195 | 7.274 | 7.298 | 7.221 | 6.879 | **7.832** |

### D.3 EXPERIMENTAL COMPARISON OVER LATENT EXPLORATION

We compare our methods AdvLatGAN-z with two representative works Evolgan and DDLS that combine latent exploration and GANs. For Evolgan implementation, we choose four sets of hyper-parameters for evaluation. Note that we did not find the official codes for Evolgan and DDLS for real-world experiment, so we reproduce the methods according to their papers. The experimental setting is aligned with Table 4 and we evaluate on CIFAR-10 with four framworks: DCGAN, WGAN, WGANGP and SNGAN. We conduct AdvLatGAN-z on a 10000 generated images batch and then evaluate the results. Inception Score and Fréchet Inception Distance are adopted as the evaluation metrics. The experimental results are presented in Table 9 and Table 10.

### D.4 EXPERIMENTAL COMPARISON OVER GAN MODEL WITH ADVERSARIAL TRAINING

We compare our methods AdvLatGAN-qua and AdvLatGAN-qua+ with two representative works ASGAN and Robust GAN training that combine adversarial training and GANs. These two methods are related to perturbations on the generated samples and the real samples respectively. The experimental setting is aligned with Table 4 and we evaluate on CIFAR-10 with four framworks: DCGAN,

Table 10: Experimental comparisons over GAN with adversarial training evaluated by **Fréchet Inception Distance**.

| Framework | Evolgan (a=0,b=80) | Evolgan (a=0.5,b=80) | Evolgan (a=0.5,b=20) | Evolgan (a=1,b=20) | DDLS | AdvLatGAN-z |
|---|---|---|---|---|---|---|
| DCGAN | 49.237 | 48.428 | 49.080 | 48.945 | 58.133 | **45.763** |
| WGAN | 34.685 | 34.650 | 34.564 | 34.565 | 221.367 | **32.854** |
| WGAN-GP | 27.004 | 26.252 | 26.497 | 26.492 | 32.516 | **22.870** |
| SNGAN | 27.661 | 27.418 | 27.542 | 27.542 | 27.542 | **23.531** |

Table 11: Experimental comparisons over GAN with adversarial training evaluated by **Inception Score**.

| Framework | ASGAN | RobDis | AdvLatGAN-qua | AdvLatGAN-qua+ |
|---|---|---|---|---|
| DCGAN | $6.21 \pm 0.07$ | $6.03 \pm 0.03$ | $6.28 \pm 0.04$ | $\mathbf{6.58 \pm 0.34}$ |
| WGAN | $4.10 \pm 0.01$ | $6.84 \pm 0.04$ | $7.21 \pm 0.04$ | $\mathbf{7.76 \pm 0.07}$ |
| WGAN-GP | $6.80 \pm 0.02$ | $7.41 \pm 0.06$ | $7.60 \pm 0.06$ | $\mathbf{8.59 \pm 0.10}$ |
| SNGAN | $7.21 \pm 0.03$ | $0.03 \pm 0.06$ | $7.58 \pm 0.03$ | $\mathbf{8.13 \pm 0.06}$ |

Table 12: Experimental comparisons over GAN with adversarial training evaluated by **Fréchet Inception Distance**.

| Framework | ASGAN | RobDis | AdvLatGAN-qua | AdvLatGAN-qua+ |
|---|---|---|---|---|
| DCGAN | $45.3 \pm 0.4$ | $41.9 \pm 0.4$ | $41.7 \pm 1.0$ | $\mathbf{40.1 \pm 1.5}$ |
| WGAN | $89.7 \pm 1.3$ | $1.3 \pm 0.4$ | $27.3 \pm 0.7$ | $\mathbf{27.2 \pm 0.5}$ |
| WGAN-GP | $32.3 \pm 0.3$ | $24.5 \pm 0.1$ | $22.6 \pm 0.4$ | $\mathbf{18.3 \pm 1.1}$ |
| SNGAN | $26.7 \pm 0.5$ | $50.9 \pm 2.7$ | $22.3 \pm 0.5$ | $\mathbf{21.9 \pm 0.3}$ |

WGAN, WGANGP and SNGAN. Inception Score and Fréchet Inception Distance are adopted as the evaluation metrics. The experimental results are presented in Table 11 and Table 12.

## E    DETAILS OF LATENT SAMPLING IMPROVEMENT EXPERIMENT

### E.1    PROTOCOLS FOR SYNTHETIC EXPERIMENTS.

We simulate two synthetic datasets. Ring dataset is a mixture of 8 2-D Gaussians $p(\mathbf{z})$ with mean $\{(2\cos(i\pi/4), 2\cos(i\pi/4))\}_{i=1}^{8}$ and standard deviation 0.001. 12.5K samples are simulated from each Gaussian distribution. 50K samples from $p(\mathbf{z})$ are used to generate $\mathbf{x}$ for test. Grid dataset is a mixture of 25 2-D isotropic Gaussians i.e. $p(\mathbf{z})$ with mean $\{(2i, 2j)\}_{i,j=-2}^{2}$ and standard deviation 0.0025. 4K samples are simulated from each Gaussian. 20K samples from $p(\mathbf{z})$ are used to generate target samples $\{\tilde{\mathbf{x}}\}$ for test. In the synthetic experiments, we apply fully-connected networks for generation and the architectures are shown in Table 13 and Table 14.

<table>
<tr><td colspan="3">Table 13: Architecture of generator $G$.</td><td colspan="3">Table 14: Architecture of discriminator $D$.</td></tr>
<tr><td>Layer</td><td>Output size</td><td>Activation</td><td>Layer</td><td>Output size</td><td>Activation</td></tr>
<tr><td>Linear</td><td>100</td><td>ReLu</td><td>Linear</td><td>100</td><td>ReLu</td></tr>
<tr><td>Linear</td><td>200</td><td>ReLu</td><td>Linear</td><td>200</td><td>ReLu</td></tr>
<tr><td>Linear</td><td>100</td><td>ReLu</td><td>Linear</td><td>100</td><td>ReLu</td></tr>
<tr><td>Linear</td><td>2</td><td>-</td><td>Linear</td><td>1</td><td>-</td></tr>
</table>

### E.2    DETAILS FOR RING SYNTHESIZING EXPERIMENTS.

To better demonstrate the practical effect and to show the characteristics of the network during the training process, we choose 2000 as the total number of iterations (1 epoch), and the result is shown in the left picture in Fig. 10. We apply the latent transform technique for the network in the left picture: frozen all parameters of the neural network and iterating 20,000 points sampled in a 2D-Gaussian distribution under the constraint of L-infinity norm. The step size is set as 0.003 and its optimizing target is upgrading discriminator judged score. After iterating 8,000 times, We get the high-quality

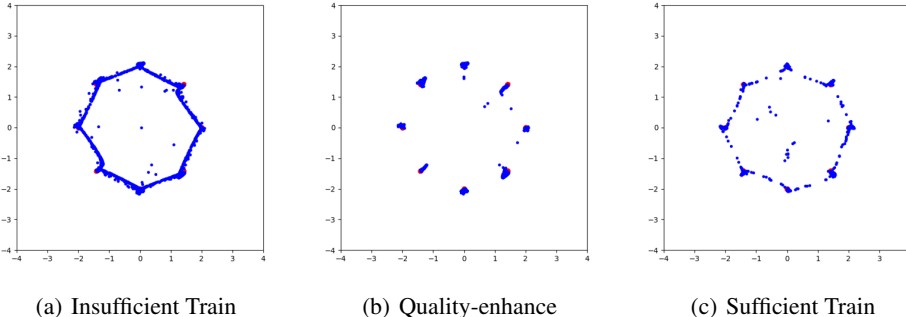

| (a) Insufficient Train | (b) Quality-enhance | (c) Sufficient Train |

Figure 10: Left: points generated by the generator after training for one epoch. Middle: apply latent iterative sampling strategy for the generator shown in the left image. Right: generated points after training 100 epochs.

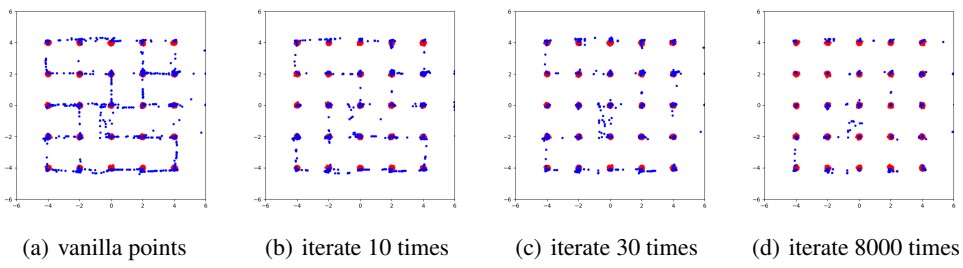

| (a) vanilla points | (b) iterate 10 times | (c) iterate 30 times | (d) iterate 8000 times |

Figure 11: Results of using AdvLatGAN-qua.

samples in Fig. 10 (shown in the middle) which outperform the results after training 100 epochs. It is a powerful evidence to prove the problem of imbalance in the input latent space and that discriminator tends to converge faster.

### E.3 DETAILS FOR GRID SYNTHESIZING EXPERIMENTS.

We try to simulate the case that the inappropriate selection of network architecture to prove the application prospects of latent transform in this situation. All hyperparameters are consistent with the previous ring experiment besides choosing training 50 epochs to make sure the network is fully trained. Because of the inappropriate choosing model architecture, the training result is terrible. However, by applying with latent transform policy, almost all points converge towards target points. The result is shown in Fig. 10.

### E.4 DETAILS FOR MNIST EXPERIMENT.

The DCGAN model and WGAN-GP model are used in this experiment. We use DCGAN model for generating images of CIFAR-10 and ResNet model for STL-10 dataset. The hyperparameters and model structures are consistent with that used in https://github.com/w86763777/pytorch-gan-collections with a total of 30 iterations, each with a step size of 0.03 and a batch size of 8000. The configuration of the trained DCGAN model: batch size is 128, D-learning rate and G-learning rate are both 0.0002, the loss function is bce loss, the discriminator updates one step per generator iteration, the latent dimension is 2, and the total training step is 20,000. The configuration of training WGAN-GP model: the batch size is 128, the D-learning rate and G-learning rate are both 0.0002, the parameters of discriminator updates one step per generator iteration, the latent dimension is 2, and the total training step is 50,000.

**Details for CIFAR-10 and STL-10 Experiment.** The SNGAN and WGAN-GP are used in this experiment, referencing https://github.com/w86763777/pytorch-gan-collections with a total of 100 iterations, each with step size of 0.01 and the batch size is set as 1500 for STL-10 and 3000 for CIFAR-10.

Original    Quality Enhanced   Quality Weakened

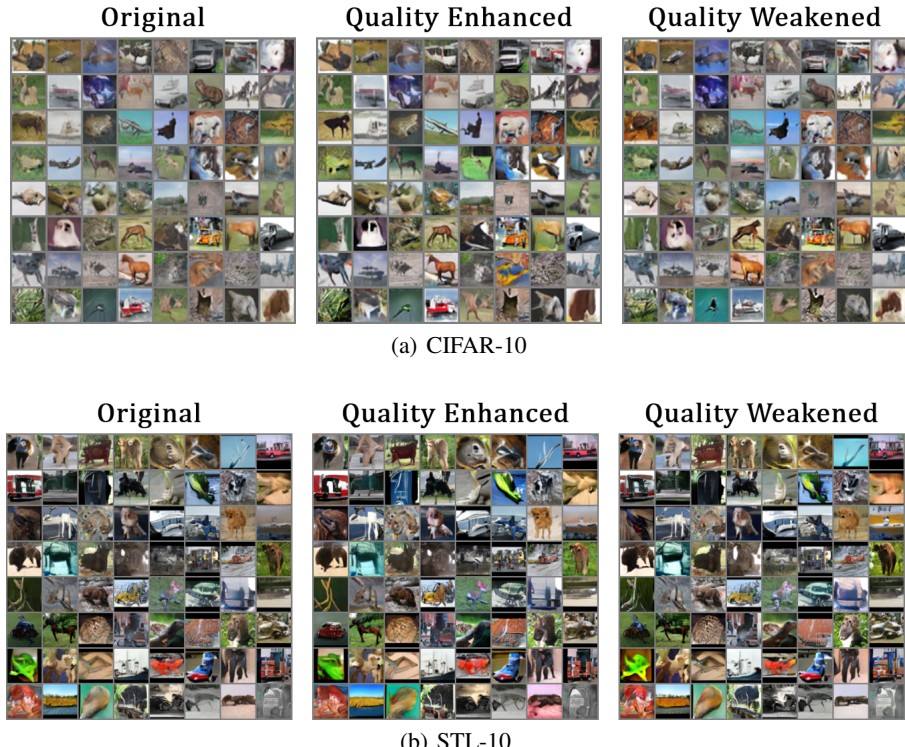

(a) CIFAR-10

Original    Quality Enhanced   Quality Weakened

(b) STL-10

Figure 12: Results of SNGAN on CIFAR-10 and STL-10.

The configuration of SNGAN for CIFAR-10: the batch size is 128, the D-learning rate and G-learning rate are both 0.0002, the loss function is hinge loss, the discriminator updates one step per generator iteration, the latent dimension is 100, and the total training step is 100,000. The training configuration of the trained WGAN-GP model for CIFAR-10: the loss function is Wasserstein loss and other configurations are the same as SNGAN.

The training configuration of SNGAN model for STL-10: the batch size is 64, the D-learning rate and G-learning rate are both 0.0002, the loss function is hinge loss, the discriminator updates 5 steps per generator iteration, the latent dimension is 128, and the total training step is 100,000. The training configuration of the trained WGAN-GP model for CIFAR-10: the loss function is Wasserstein loss and other configurations are the same with SNGAN. The results are shown in Fig. 12, Fig. 13.

## F EXPERIMENT FOR ADVERSARIAL SAMPLE SEARCH FOR DIVERSITY

We propose AdvLatGAN-div to improve the mapping property for diversity, which introduces the latent $\mathbf{z}$ iterations by Eq. 8 into the original bi-level optimization process. The validity of the algorithm requires the effectiveness of the $\mathbf{z}$ iterations searching for samples with varying diversity performance guided by Eq. 8, which we verify by experiments.

**Details for CIFAR-10 Experiment.** We first find a random latent vector of batch size 64, respectively do multiple-step gradient sign descent and multiple-step gradient sign ascent on the ratio of the distance between the original image and the target image to the distance of original input vector and target input vector (which denoted as Eq. 8). The results are presented in Fig. 14. The number of iterative steps is set to 300 and the step length is 0.03. The model is a pre-trained CIFAR-10 DCGAN model.

**Details for FACADES Experiment.** Under the condition of the same image input, 20 latent space vectors are randomly generated to complete the generation, and each vector is updated in two directions to obtain 20 sets of comparison images.

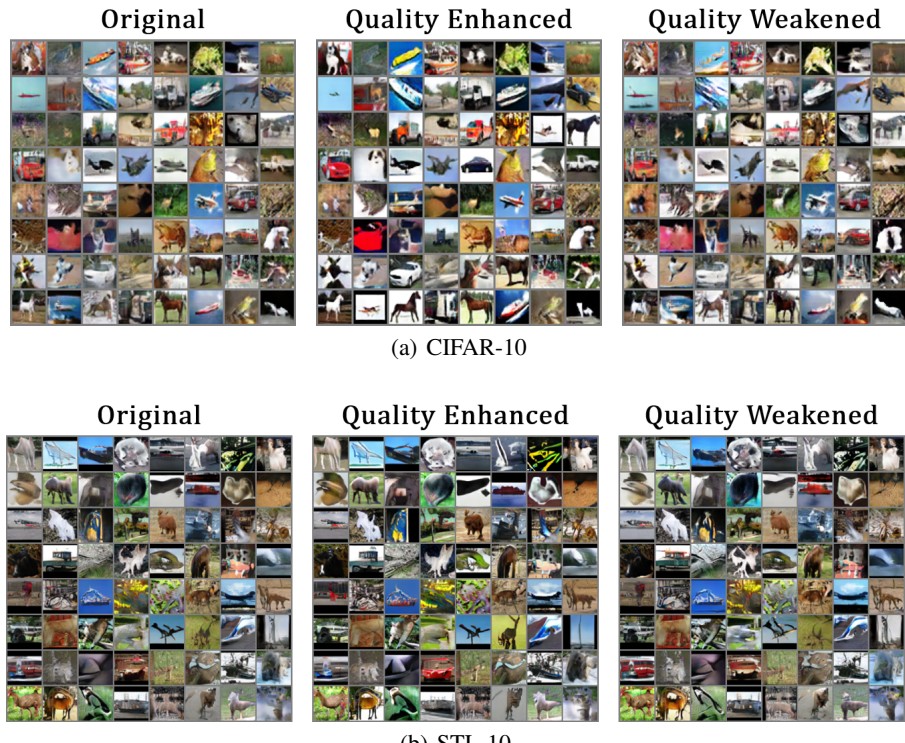

(a) CIFAR-10

(b) STL-10

Figure 13: Results of WGAN-GP on CIFAR-10 and STL-10.

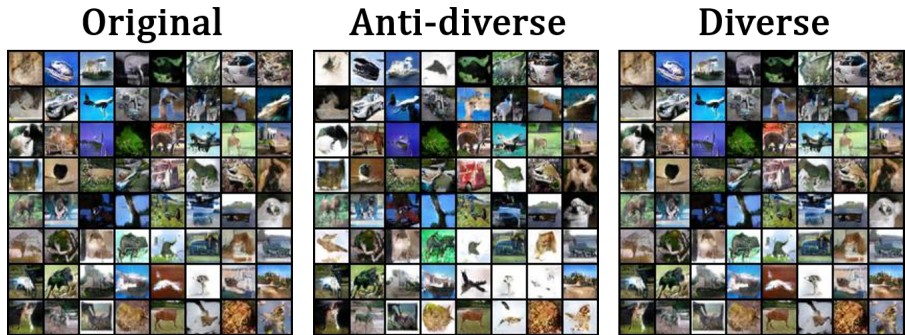

Figure 14: Results of latent iterations for diversity by using DCGAN Radford et al. (2016) pre-trained on CIFAR-10.

# G   DETAILS OF ADVERSARIAL LATENT GAN EXPERIMENT

## G.1   EXPERIMENT FOR ITERATIVE QUALITY IMPROVEMENT

### G.1.1   DETAILS FOR CIFAR-10 EXPERIMENT

We regard latent space variables as iterable parameters and in case of divergence, we apply learning rate $\epsilon$ for latent vector updating process. And the hyperparameters we use in this experiment are shown in Table 17. We use basic convolutional neural networks for test. The architectures are presented in Table 15 and Table 16.

| learning rate $\epsilon$ | DCGAN | WGAN | WGAN-GP | SNGAN |
|---|---|---|---|---|
| AT+GD | 0.006 | 0.0008 | 0.001 | 0.005 |
| AT+LIN | 0.01 | 0.003 | 0.005 | 0.02 |

Table 17: The hyperparameter setting in CIFAR-10 adversarial training experiment.

| learning rate $\epsilon$ | DCGAN | WGAN | WGAN-GP | SNGAN |
|---|---|---|---|---|
| AT+LIN | 0.01 | 0.003 | 0.005 | 0.02 |

Table 18: Hyperparameter setting in adversarial training Experiment on STL-10.

Table 15: Architecture of generator $G$ in CIFAR10 experiment.

| Layer | Output size |
|---|---|
| ConvTranspose2d | $2 \times 2 \times 1024$ |
| BatchNorm2d | $2 \times 2 \times 1024$ |
| Relu | $2 \times 2 \times 1024$ |
| ConvTranspose2d | $4 \times 4 \times 512$ |
| BatchNorm2d | $4 \times 4 \times 512$ |
| Relu | $4 \times 4 \times 512$ |
| ConvTranspose2d | $8 \times 8 \times 256$ |
| BatchNorm2d | $8 \times 8 \times 256$ |
| Relu | $8 \times 8 \times 256$ |
| ConvTranspose2d | $16 \times 16 \times 128$ |
| BatchNorm2d | $16 \times 16 \times 128$ |
| Relu | $16 \times 16 \times 128$ |
| ConvTranspose2d | $32 \times 32 \times 3$ |
| Tanh | $32 \times 32 \times 3$ |

Table 16: Architecture of discriminator $D$ in CIFAR10 experiment.

| Layer | Output size |
|---|---|
| Conv2d | $16 \times 16 \times 64$ |
| LeakyRelu | $16 \times 16 \times 64$ |
| Conv2d | $8 \times 8 \times 128$ |
| LeakyRelu | $8 \times 8 \times 128$ |
| BatchNorm2d | $8 \times 8 \times 128$ |
| Conv2d | $4 \times 4 \times 256$ |
| LeakyRelu | $4 \times 4 \times 256$ |
| BatchNorm2d | $4 \times 4 \times 256$ |
| Conv2d | $2 \times 2 \times 512$ |
| LeakyRelu | $2 \times 2 \times 512$ |
| BatchNorm2d | $2 \times 2 \times 512$ |
| faltten | 2048 |
| linear | 1 |

Table 19: Architecture of generator $G$ in STL-10 experiment.

| Layer | Output size |
|---|---|
| ConvTranspose2d | $3 \times 3 \times 1024$ |
| BatchNorm2d | $3 \times 3 \times 1024$ |
| Relu | $3 \times 3 \times 1024$ |
| ConvTranspose2d | $6 \times 6 \times 512$ |
| BatchNorm2d | $6 \times 6 \times 512$ |
| Relu | $6 \times 6 \times 512$ |
| ConvTranspose2d | $12 \times 12 \times 256$ |
| BatchNorm2d | $12 \times 12 \times 256$ |
| Relu | $12 \times 12 \times 256$ |
| ConvTranspose2d | $24 \times 24 \times 128$ |
| BatchNorm2d | $24 \times 24 \times 128$ |
| Relu | $24 \times 24 \times 128$ |
| ConvTranspose2d | $48 \times 48 \times 3$ |
| Tanh | $48 \times 48 \times 3$ |

Table 20: Architecture of discriminator $D$ in STL-10 experiment.

| Layer | Output size |
|---|---|
| Conv2d | $24 \times 24 \times 64$ |
| LeakyRelu | $24 \times 24 \times 64$ |
| Conv2d | $12 \times 12 \times 128$ |
| LeakyRelu | $12 \times 12 \times 128$ |
| BatchNorm2d | $12 \times 12 \times 128$ |
| Conv2d | $6 \times 6 \times 256$ |
| LeakyRelu | $6 \times 6 \times 256$ |
| BatchNorm2d | $6 \times 6 \times 256$ |
| Conv2d | $3 \times 3 \times 512$ |
| LeakyRelu | $3 \times 3 \times 512$ |
| BatchNorm2d | $3 \times 3 \times 512$ |
| faltten | 4608 |
| linear | 1 |

## G.2 DETAILS FOR STL-10 EXPERIMENT.

The learning rates for updating latent variables in STL-10 experiment are presented in Table 18. We use basic convolutional neural networks for test. The architectures are presented in Table 19 and Table 20,

### G.3 EXPERIMENT FOR ITERATIVE DIVERSITY IMPROVEMENT

#### G.3.1 DETAILS FOR CIFAR-10 EXPRIMENT

The experiment is obtained based on the experimental extension of MSGAN (Mao et al., 2019), and the operation of latent variable search is added to the original design. The latent vector is updated in the experiment guided by Eq. 8 with a gradient update strategy constrained by $\ell_\infty$. In our experimental setting, we adopt 3 updates per iteration and the step size is set as 0.01. Other training parameters are referred to the original MSGAN experiment `https://github.com/HelenMao/MSGAN` without modification. The FID and JSD metrics for each class are calculated based on 5000 images generated for each class, and the total FID and JSD metrics are calculated by the collection of generated images for each class mentioned above.

#### G.3.2 DETAILS FOR STL-10 EXPRIMENT

The basic setup of this experiment follows the CIFAR-10 experiment, and one difference is that: we use the supervised test part of STL10 as the real dataset for training and calculate the JSD and FID metrics for the generated images and the labeled training part of STL10.

