# OpenReview forum: "Improving Generative Adversarial Networks via Adversarial Learning in Latent Space"
_ICLR.cc/2022/Conference — ICLR 2022 Submitted_

### Official Review · Reviewer_JVu6 · 2021-11-02

**Correctness:** 3
**Technical Novelty And Significance:** 2
**Empirical Novelty And Significance:** 2
**Recommendation:** 6
**Confidence:** 4

**Main Review:**

1.    Related works have not been cited. For example, the following paper performs a similar optimization technique with a different objective.

a.       Yan Wu, Jeff Donahue, David Balduzzi, Karen Simonyan, Timothy Lillicrap. LOGAN: Latent Optimisation for Generative Adversarial Networks. (https://arxiv.org/pdf/1912.00953.pdf)

b. https://arxiv.org/abs/2005.02435

c. https://arxiv.org/abs/1809.03627

Please do a thorough literature review to incorporate any such missing work.

2.       The authors propose five methods viz (i) AdvLatGAN-z, (ii) AdvLatGAN-qua, (iii) AdvLatGAN-qua+, (iv) AdvLatGAN-div, and (v) AdvLatGAN-div+.

Each method either focuses on improving quality or aims to enhance diversity.

It would be interesting to see what happens when the proposed objectives in equation (5) and equation (8) are combined. Does the hybrid method outperform both AdvLatGAN-qua+ and AdvLatGAN-div+?

3.   Another limitation of the work is that they compute FID and JSD. FID, although a widely used metric, is not able to quantify the quality and diversity as it is a unidimensional score. Therefore, it would be nice to quantitatively verify the claims of enhancement in quality and diversity in AdvLatGAN-qua+ and AdvLatGAN-div+ respectively. Comparing other metrics such as precision/recall or density/coverage will be more meaningful towards such goals.

a.  Density/Coverage: Muhammad Ferjad Naeem, Seong Joon Oh, Youngjung Uh, Yunjey Choi, and Jaejun Yoo. Reliable fidelity and diversity metrics for generative models. In International Conference on Machine Learning, 2020. (https://arxiv.org/abs/2002.09797)


b.   Precision/Recall: Mehdi S. M. Sajjadi, Olivier Bachem, Mario Lucic, Olivier Bousquet, Sylvain Gelly. Assessing Generative Models via Precision and Recall. In NIPS, 2018. (https://arxiv.org/abs/1806.00035)

4. The number of baselines used in the experiment section is too less for the CIFAR-10 case?

5. It would be good have experiments on large-scale datasets such as FFHQ.

6. I wonder if FGSM is the only method this can be applied with or any other can be used as well? If so, how does the method depend on the adversarial training method employed?

7.  I am not sure why MSGAN was chosen as the baseline for regularization?

8. There is no theoretical justification on why should the proposed method work? There is some empirical evidence but there would be better to have some theoretical backing on why this method should aid in avoiding mode-collapse.


**Summary Of The Paper:**

The authors demonstrate that the generator in a GAN is a continuous function two latent codes that are close in the latent space are mapped to two images that are close in the pixel space. However, the quality of the generated images is not preserved as quality is not a continuous function in pixel space. To address this issue, the authors propose to transform the original latent codes and demonstrate that it results in better generation quality and diversity.

**Summary Of The Review:**

Please refer to the above comments.

---

> ### Author Response · Authors · 2021-11-20
> **Response to Reviewer JVu6**
>
> Thank you for the detailed feedback. Here are our responses to your comments:
>
> **Q1: Related works.**
>
> **R1:** Thanks for sharing these related works. We will include these and other related works in our main paper to strengthen the sufficiency of the related work section.
>
> **Q2: It would be interesting to see what happens when the proposed objectives in equation (5) and equation (8) are combined.**
>
> **R2:** Thanks for your valuable thoughts. First, it is worth noting that AdvLatGAN-div is built on the conditional setting, thus the combination of the two methods can only works in this conditional scenario, which we consider will limit our contributions to some extent. Second, we actually have already made efforts to promote the combination, but this does not work as efficiently as the two algorithms do separately. To explain this, We believe that there may exist a trade-off between generation quality and diversity, and the combined optimization objective may be too complex to effectively solve.
>
> **Q3: Metric separating the fidelity evaluation and the diversity evaluation.**
>
> **R3:** Thanks for the valuable suggestion which is more than suitable for our setting. In the rebuttal phrase, we re-evaluate some of our experimental results with the density/coverage [1] metric as you recommended. Note that the concept of *quality* in our work differs from *fidelity* in [1]. *Quality* here stems from the judge of the discriminator which can be more precisely adopted as an integrated concept containing fidelity and diversity, which describes the matching degree between the generated distribution and real distribution. Thus here we do not separate the two metrics to evaluate AdvLatGAN-qua and AdvLatGAN-div but adopt them both in each setting to show the effectiveness in a more comprehensive manner. Here we additionally compare to [2] and [3] which interplay between GANs and adversarial learning. We present our experimental results below:
>
> Results over coverage metric on CIFAR10 with DCGAN, WGAN, WGAN-GP and SNGAN frameworks:
>
> |        | vanilla | ASGAN [1] | RobDis [2] | AdvLatGAN-qua | AdvLatGAN-qua+ |
> | ------ | ------- | -------- | --------- | ------------- | -------------- |
> | DCGAN  | 0.376   | 0.405    | 0.414     | 0.399         | **0.421**      |
> | WGAN   | 0.443   | 0.163    | 0.470     | 0.505         | **0.530**      |
> | WGANGP | 0.562   | 0.458    | 0.564     | 0.590         | **0.667**      |
> | SNGAN  | 0.547   | 0.513    | 0.358     | 0.566         | **0.601**      |
>
>
> Results over density metric on CIFAR10:
>
> |        | vanilla | ASGAN [1] | RobDis [2] | AdvLatGAN-qua | AdvLatGAN-qua+ |
> | ------ | ------- | -------- | --------- | ------------- | -------------- |
> | DCGAN  | 0.354   | 0.372    | **0.375** | 0.368         | 0.345          |
> | WGAN   | 0.384   | 0.262    | 0.391     | **0.414**     | 0.407          |
> | WGANGP | 0.462   | 0.362    | 0.462     | 0.495         | **0.530**      |
> | SNGAN  | 0.440   | 0.418    | 0.417     | 0.444         | **0.456**      |
>
> Here we also present results of Inception Score and Fréchet Inception Distance here.
>
> Results of Inception Score metric on CIFAR10:
>
> |        | vanilla | ASGAN [1] | RobDis [2] | AdvLatGAN-qua | AdvLatGAN-qua+ |
> | ------ | ------- | -------- | --------- | ------------- | -------------- |
> | DCGAN  | 5.92    | 6.03     | 6.28      | 6.21          | **6.58**       |
> | WGAN   | 6.63    | 4.10     | 6.84      | 7.21          | **7.76**       |
> | WGANGP | 7.47    | 6.80     | 7.41      | 7.60          | **8.59**       |
> | SNGAN  | 7.29    | 7.21     | 6.30      | 7.58          | **8.13**       |
>
> Results of Fréchet Inception Distance metric on CIFAR10:
>
> |        | vanilla | ASGAN[1] | RobDis[2] | AdvLatGAN-qua | AdvLatGAN-qua+ |
> | ------ | ------- | ----- | ------ | ------------- | -------------- |
> | DCGAN  | 46.4    | 45.3  | 41.9   | 41.7          | **40.1**       |
> | WGAN   | 32.8    | 89.7  | 31.4   | 27.3          | **27.2**       |
> | WGANGP | 24.7    | 32.3  | 24.5   | 22.6          | **18.3**       |
> | SNGAN  | 25.5    | 26.7  | 50.9   | 22.3          | **21.9**       |
>
>
> We find the results of the density metric on DCGAN setting not that delightful while the performance of the other three metrics with the same setting is pretty satisfying. We believe this may stem from the departure of evaluating over the matching degree of the whole distribution which can not reflect the whole aspects of models' performance. The divided density metric may happen to target at the dimension in which AdvLatGAN-qua+ are weak.

---

> > ### Author Response · Authors · 2021-11-20
> > **Response to Reviewer JVu6 (Cont.)**
> >
> > We also re-evaluate results in Table 6 and Table 7 on density/coverage metrics.
> >
> > Results on CIFAR-10:
> >
> > | Metrics  | Models         | overall   | airplane   | automobile | bird      | cat       | deer      | dog       | frog      | horse     | ship      | truck     |
> > | -------- | -------------- | --------- | ---------- | ---------- | --------- | --------- | --------- | --------- | --------- | --------- | --------- | --------- |
> > | coverage | MSGAN          | 0.568     | 0.651      | 0.690      | 0.460     | 0.674     | 0.736     | 0.452     | 0.776     | **0.779** | 0.817     | **0.612** |
> > | coverage | AdvLatGAN-div+ | **0.726** | **0.677**  | **0.752**  | **0.464** | **0.748** | **0.797** | **0.466** | **0.843** | 0.743     | **0.877** | 0.604     |
> > | density  | MSGAN          | 0.350     | **0.338** | 0.216      | **0.308** | 0.635     | **0.532** | 0.272     | 0.665     | 0.296     | 0.331     | 0.224     |
> > | density  | AdvLatGAN-div+ | **0.516** | 0.333      | **0.255**  | 0.267     | **0.679** | 0.511     | **0.318** | **0.674** | **0.300** | **0.353** | **0.231** |
> >
> > Results on STL-10:
> >
> > | Metrics  | Models         | overall   | airplane  | bird      | car       | cat       | deer      | dog       | house     | monkey    | ship      | truck     |
> > | -------- | -------------- | --------- | --------- | --------- | --------- | --------- | --------- | --------- | --------- | --------- | --------- | --------- |
> > | coverage | MSGAN          | 0.361     | 0.383     | **0.364** | 0.169     | 0.418     | 0.306     | 0.289     | 0.271     | 0.280     | 0.329     | 0.226     |
> > | coverage | AdvLatGAN-div+ | **0.396** | **0.408** | 0.329     | **0.226** | **0.508** | **0.424** | **0.309** | **0.390** | **0.335** | **0.395** | **0.244** |
> > | density  | MSGAN          | 0.332     | 0.192     | 0.264     | 0.089     | 0.598     | 0.399     | 0.397     | 0.150     | 0.251     | 0.117     | 0.116     |
> > | density  | AdvLatGAN-div+ | **0.443** | **0.235** | **0.328** | **0.111** | **0.733** | **0.456** | **0.429** | **0.194** | **0.353** | **0.165** | **0.126** |
> >
> > The results show the effectiveness of AdvLatGAN-div+. The improvements on the overall data evaluation reach at most 27.8% in coverage metric and 47.4% in density metric.
> >
> > **Q4: The number of baselines used in the experiment section is too less for the CIFAR-10 case?**
> >
> > **R4:** We supplement more baselines LSGAN [4], WGAN-div [5] and ACGAN [6] for comparison in CIFAR-10 evaluation setting, along with LSGAN and WGAN-div in STL-10 evaluation setting (ACGAN does not work on unlabeled STL-10). We present the results below:
> >
> > Results over Inception Score metric on CIFAR-10:
> >
> > | model    | vanilla        | AdvLatGAN-qua   | AdvLatGAN-qua+      |
> > | -------- | -------------- | --------------- | ------------------- |
> > | LSGAN    | 5.87$\pm$ 0.14 | 6.13 $\pm$ 0.27 | **6.38 $\pm$ 0.18** |
> > | WGAN-div | 7.43$\pm$ 0.02 | 7.81 $\pm$ 0.03 | **8.67 $\pm$ 0.11** |
> > | ACGAN    | 6.02$\pm$ 0.28 | 6.06 $\pm$ 0.21 | **6.21 $\pm$ 0.30** |
> >
> > Results over Fréchet Inception Distance metric on CIFAR-10:
> >
> > | model    | vanilla        | AdvLatGAN-qua  | AdvLatGAN-qua+     |
> > | -------- | -------------- | -------------- | ------------------ |
> > | LSGAN    | 49.3 $\pm$ 2.2 | 42.8 $\pm$ 1.3 | **41.0 $\pm$ 1.3** |
> > | WGAN-div | 23.8 $\pm$ 0.3 | 20.6 $\pm$ 0.4 | **16.0 $\pm$ 0.5** |
> > | ACGAN    | 59.5 $\pm$ 1.5 | 53.7 $\pm$ 1.6 | **53.1 $\pm$ 0.4** |
> >
> > Results over Inception Score metric on STL-10:
> >
> > | model    | vanilla        | AdvLatGAN-qua   | AdvLatGAN-qua+       |
> > | -------- | -------------- | --------------- | -------------------- |
> > | LSGAN    | 7.08$\pm$ 0.12 | 7.16 $\pm$ 0.15 | **7.55 $\pm$ 0.18**  |
> > | WGAN-div | 8.82$\pm$ 0.02 | 9.00 $\pm$ 0.01 | **10.68 $\pm$ 0.17** |
> >
> > Results over Fréchet Inception Distance metric on STL-10:
> >
> > | model    | vanilla        | AdvLatGAN-qua  | AdvLatGAN-qua+     |
> > | -------- | -------------- | -------------- | ------------------ |
> > | LSGAN    | 62.9 $\pm$ 2.2 | 58.5 $\pm$ 1.3 | **57.1 $\pm$ 1.5** |
> > | WGAN-div | 37.7 $\pm$ 0.2 | 32.0 $\pm$ 0.6 | **24.3 $\pm$ 1.0** |
> >
> >
> > **Q5: I wonder if FGSM is the only method this can be applied with or any other can be used as well? If so, how does the method depend on the adversarial training method employed?**
> >
> > **R5:** Actually we have tried PGD and adversarial iteration under $\ell_{2}$ constraint and w/o constraint, among which FGSM achieves the best results. Other methods can achieve improvements, but they are not as significant as FGSM.
> >
> > The adversarial training methods are AdvLatGAN-qua and AdvLatGAN-div, whose implementation details are mostly presented in Algorithm 1 and Algorithm 2. The changes over the original training process can be viewed as adversarial training (hard sample mining) respectively against the discriminator and the generator under the MS regularization (see Sec.3.2 and Sec.3.3 for the detailed illustration).

---

> > > ### Author Response · Authors · 2021-11-20
> > > **Response to Reviewer JVu6 (Cont.)**
> > >
> > > **Q6: Why MSGAN was chosen as the baseline for regularization.**
> > >
> > > **R6:** AdvLatGAN-div introduces adversarial learning for hard sample search into MSGAN (see Sec.3.3 for a more detailed illusration), thus comparison to MSGAN is natural and can show the effectiveness in a direct manner.
> > >
> > > We hope this response could help address your concerns, and wish to receive your further feedback soon.
> > >
> > > ---
> > > **References:**
> > >
> > > [1] Reliable fidelity and diversity metrics for generative models. ICML 2020.
> > >
> > > [2] Don't let your Discriminator be fooled. ICLR 2018.
> > >
> > > [3] Adversarial symmetric GANs: Bridging adversarial samples and adversarial networks. Neural Networks 2021.
> > >
> > > [4] Least squares generative adversarial networks. ICCV 2017.
> > >
> > > [5] Wasserstein divergence for gans. ECCV 2018.
> > >
> > > [6] Conditional image synthesis with auxiliary classifier gans. ICML 2017.

---

### Official Review · Reviewer_uSqk · 2021-11-03

**Correctness:** 4
**Technical Novelty And Significance:** 3
**Empirical Novelty And Significance:** 3
**Recommendation:** 6
**Confidence:** 3

**Main Review:**

Pros:
1. The paper is well organized, with clear sub-titles and clear logic flow;
2. Ablation study on various baseline GAN architecture (DCGAN, WGAN, WGAN-GP, SNGAN) is conducted to show generalizability of such sampling method.

Cons:
1. Compare with baseline method MSGAN: (1) why it only compare the div+ with MSGAN; (2) improvement over baseline method MSGAN is very limited.
2. If generator trained with better quality regularization, the latent space after mapping should have better continuity? Comparison like Fig 3 after training would be needed to prove that.

Some minor issues:
1. The paper is a bit redundant on algorithms and figures. For example,  it seems lengthy to include both algorithm 1 and 2 in the main paper
2. Fig 1. It’s not obvious what the latent space did in this figure


**Summary Of The Paper:**

This work proposed a sample shifting method in GAN, formulated as adding intermediate latent space to generated pixel space. Such method is based on observation of continuous mapping limit: image quality in pixel space is not as continuous as latent space; limited latent space will incur mode collapse thus poor image diversity. The main contributions are: a new optimization problem as sampling method to improve image generation by quality and diversity, propose to use I-FGSM optimization method to achieve this sampling optimization problem. The experiment showed improvement on public dataset of STL-10, CIFAR-10.

**Summary Of The Review:**

The paper is overall well written, and the idea is very clear and elegant. Meanwhile, I have some doubt on the sufficiency of experiment to show the improvement from quality. Also the improvement over baseline method MSGAN seems minor for me. Expect the rebuttal to clear my doubt.

---

> ### Author Response · Authors · 2021-11-20
> **Response to Reviewer uSqk**
>
> Thank you for the detailed feedback. Here are our responses to your comments:
>
> **Q1: Compare with baseline MSGAN: (1) why it only compares the div+ with MSGAN; (2) improvement over baseline method MSGAN is very limited.**
>
> **R1:** (1) We found in the experiments that AdvLatGAN-z can largely affect JSD results, which is designed to target at diversity. According to our illustration in Sec.1, AdvLatGAN-z will transform to avoid sampling for bad generation, thus it can be natural to cause a diversity loss. This can be reflected in JSD evaluation which may confuse the readers to doubt the effectiveness of the method. In fact the extra boost compared to div+ on JSD is an overflow boost, what we need is the evaluation of meaningful samples. So we choose div+ for comparison.
>
> (2) We understand your concern as the improvement achieved in the diversity setting may not be as significant as in the quality setting. However, as in experiments where the models are trained with the overall dataset, it can be difficult to achieve remarkable improvement in each label since there may exist trade-offs between the optimizations for different data, and the optimization considering the overall data may not benefit the performance in each label. In fact, for the overall evaluation results (see Table 6 and Table 7), AdvLatGAN-div+ has achieved at most 11.5% improvement in FID, 36.0% improvement in JSD, 47.4% in density and 27.8% in coverage, which we believe is significant enough.
>
> **Q2: If generator trained with better quality regularization, the latent space after mapping should have better continuity? Comparison like Fig 3 after training would be needed to prove that.**
>
> **R2:** The training algorithm AdvLatGAN-qua aims to achieve better mapping properties (see Fig. 2 for the schematic diagram) and the bottleneck from the continuous latent distribution and continuous mapping function is solved by AdvLatGAN-z. In other words, AdvLatGAN-qua has nothing to do with continuity. And the effectiveness of AdvLatGAN-qua on improving the mapping can be demonstrated by experimental results in Sec.4.3.
>
> For solving the Fig. 3 issue, to show the effectiveness of AdvLatGAN-z, we experiment on invalid samples with the same setting as in Fig. 3. Note that AdvLatGAN-z works at the sample level (or distribution level as a distribution shift characterized by sample transforms) in the latent space. In the Fig. 3 setting, when the sampling encounters invalid examples, AdvLatGAN-z is able to transform the bad $z$ to $z^*(z)$ with a better ability to avoid bad generation (see Q4 for a more specific illustration). Here we verify whether invalid samples can be improved with AdvLatGAN-z.
>
> We present the results in Fig. 9. See the caption for a detailed illustration. As you may notice there appears unexpected background variation in the third column, we believe this may stem from that the defect of the image is not significant enough for the discriminator, making the optimization less on target.
>
> **Q3: The paper is a bit redundant on algorithms and figures. For example, it seems lengthy to include both algorithm 1 and 2 in the main paper.**
>
> **R3:** Thank you for the suggestion. We have edited our paper version by adding new experimental results in the main paper and moving the specific algorithms to the appendix. Since the algorithms actually present the implementation details for our training strategies which may also matter a lot, we present them at the very beginning of the appendix.
>
> **Q4: It’s not obvious what the latent space did in Fig. 1.**
>
> **R4:** We are sorry for the lack of clarity in our presentation. Here the latent space just works the same way as previous studies on GAN and we think your concern may lie in the role of latent transform $z^*$ here. We present a more detailed description for $z^*$ below:
>
> Based on Fig. 3's caption, the main focus of $z^*$ is to transform bad samples in the latent space (marked as black circles) into the *inverse images* of the real manifolds under $G$ (marked in the red dashed lines). Recall that with the generative mapping $G$ fixed, only the samples inside the *inverse images* of the real manifolds shall retain good quality in generation.
>
> Specifically, the sample pair marked purple box corresponds to the situation that close samples in latent space and pixel space can differ much in quality as the pair locate across the borderline. Under $z^*$, the black circle in the purple box is pulled close to the yellow circle, where the variation is small but the generation becomes meaningful.
>
> We hope our above response could help address your concerns, and wish to receive your further feedback soon.

---

> > ### Comment · Reviewer_uSqk · 2021-11-30
> > **Rebuttal feedback**
> >
> > Thanks the author for highlighting the improvement on vanilla model improvement, it makes the contribution more solid. I appreciate the extensive add-on experiment over different architectures. As for generalizability to larger dataset (also mentioned by reviewer JVu6), do you have comment on that?

---

> > > ### Author Response · Authors · 2021-11-30
> > > **Thank you for your feedback.**
> > >
> > > Thank you for your further feedback and the recognition of our improvement efforts. For larger datasets, we have supplemented the experiment for AdvlatGAN-z on AFHQ and FFHQ to evaluate its effectiveness on larger data (see Sec. 4.2 and Fig. 9). As for the proposed training algorithms, the experiment is very time-consuming and we are sorry that we may not provide the results under the time limit. However, we believe that our current paper version is refined in the existing experiment design and the improvements are evident (at most 12%~47% improvement on different metrics and settings). We believe that our work can contribute to this area and we sincerely hope that you could reconsider your rating.

---

### Official Review · Reviewer_ua7t · 2021-11-03

**Correctness:** 2
**Technical Novelty And Significance:** 2
**Empirical Novelty And Significance:** 2
**Recommendation:** 3
**Confidence:** 4

**Main Review:**

 + The paper is generally well written and quite easy to understand. It does however mix preliminary work (I-FGSM) with the proposed contribution in 3.2.
 + The paper is able to improve vanilla GAN models using the presented objectives.
 - The paper does fully compare to prior work. Yes Table 8 highlights some of the similarities to prior work, but the main evaluation does not compare to alternative approaches that consider the interplay between GANs and adversarial attacks or latent space exploration. In order to see the efficacy of the presented method, the paper should experimentally compare to the majority of methods highlighted in Appendix C. The paper would be a lot stronger if it could show that the design choices made here are better than other attacks (i.e. perturbations on the generated or real images, instead of latent features, e.t.c.)
- The visual quality of the presented examples is somewhat underwhelming. Does the presented method work on larger GAN architectures such as StyleGAN2 or BigGAN? Does the presented method actually address the issues highlighted in Fig 3?

**Summary Of The Paper:**

The paper looks at the problem of improving the generative quality of GANs. The paper makes improvement along two dimensions: a) The paper adjust the samples distribution of GANs using adversarial attacks (I-FGSM), and thus effectively samples from a potentially multi-modal distribution. b) The paper improves the diversity of generated samples using using adversarial attacks on the mode-seeking objective of Mao et al 2019.

**Summary Of The Review:**

The paper explores an interesting idea of adding adversarial robustness into GAN training to improve latent distribution sampling and diversification. Unfortunately, the paper falls a bit short in the experimental validation of the approach, and comparison to prior approaches.

---

Post rebuttal. The rebuttal makes a good case for their final algorithm using additional results. However, I still do not see what the paper adds on top of baselines, or how the problem setup in Figure 3 (interpolation artifacts) is actually addressed. The rebuttal mentions some experimental evidence that seems to indicate latent-space sampling can helps. However, I would need to see these results in an actual paper submission for review to feel comfortable about accepting it. As is the paper seems interesting, but not ready for publication.

---

> ### Author Response · Authors · 2021-11-20
> **Response to Reviewer ua7t**
>
> Thank you for the detailed feedback, which is very valuable for us to refine our work. Here are our responses to your comments:
>
> **Q1: Mix I-FGSM contributions in 3.2.**
>
> **R1:** We are sorry if the presentation causes any confusion. Our emphasis in this section is on what ideas we base our approach design on and how we implement them. We present the specific formulation for the iteration step (Eq.6) here because it is one of the important implementation details of our method (experiments show that during the iterations, $\ell_{\infty}$ constraint (Eq.6) works better than the $\ell_{2}$ constraint and w/o constraint). Besides, we believe that citations have already been marked for I-FGSM's contributions.
>
> We hope that our explanations can ease your concern and if you have other concerns or better layout suggestions we are happy to rearrange our presentation.
>
> **Q2: Comparison to other works interplaying between GANs and adversarial attacks or latent space exploration.**
>
> **R2:** Thanks for the valuable suggestions and the experiments mentioned can be meaningful for us to strengthen our contributions. We conduct additional experiments for the comparison of related works, which is twofold:
>
> **1. Comparison to other methods interplaying between GANs and adversarial training.**
>
> We compare our methods AdvLatGAN-qua and AdvLatGAN-qua+ with two representative works ASGAN[1] and [2] that combine adversarial training and GANs. These two methods are related to perturbations on the generated samples and the real samples respectively. See Appendix C.2 for specific descriptions of the methods. Aligning with the original experimental setting, we experiment on CIFAR-10 with four frameworks: DCGAN, WGAN, WGANGP and SNGAN. Inception Score and Fréchet Inception Distance are adopted as the evaluation metrics. The experimental results are presented as follows:
>
>
> Evaluation over Inception Score:
>
> |        | vanilla   | ASGAN[1]  | RobDis[2] | AdvLatGAN-qua | AdvLatGAN-qua+ |
> | ------ | --------- | --------- | --------- | ------------- | -------------- |
> | DCGAN  | 5.92±0.05 | 6.03±0.03 | 6.28±0.04 | 6.21±0.07     | **6.58±0.34**  |
> | WGAN   | 6.63±0.05 | 4.10±0.01 | 6.84±0.04 | 7.21±0.04     | **7.76±0.07**  |
> | WGANGP | 7.47±0.09 | 6.80±0.02 | 7.41±0.06 | 7.60±0.06     | **8.59±0.10**  |
> | SNGAN  | 7.29±0.09 | 7.21±0.03 | 6.30±0.06 | 7.58±0.03     | **8.13±0.06**  |
>
> Evaluation over Fréchet Inception Distance:
>
> |        | vanilla  | ASGAN | RobDis| AdvLatGAN-qua | AdvLatGAN-qua+ |
> | ------ | -------- | -------- | --------- | ------------- | -------------- |
> | DCGAN  | 46.4±0.7 | 45.3±0.4 | 41.9±0.4  | 41.7±1.0      | **40.1±1.5**   |
> | WGAN   | 32.8±0.5 | 89.7±1.3 | 31.4±0.4  | 27.3±0.7      | **27.2±0.5**   |
> | WGANGP | 24.7±0.2 | 32.3±0.3 | 24.5±0.1  | 22.6±0.4      | **18.3±1.1**   |
> | SNGAN  | 25.5±0.3 | 26.7±0.5 | 50.9±2.7  | 22.3±0.5      | **21.9±0.3**   |
>
> Both [1] and [2]'s official implementations are based on DCGAN backbone, so they work well on DCGAN setting as expected, but still do not outperform AdvLatGAN-qua+. Both [1] and [2] have poor adaptability on other architectures and in some cases even have quality loss compared to vanilla GAN, while our methods exhibit much better adaptability. In terms of effectiveness reflected in the evaluation, AdvLatGAN-qua+ far exceeds these methods.
>
> **2. Comparison to other methods interplaying between GANs and latent space exploration.**
>
> We compare our methods AdvLatGAN-z with two representative works Evolgan[3] and DDLS[4] that combine latent exploration and GANs. See Appendix C.1 for specific descriptions of the two methods. For Evolgan implementation, we choose four sets of hyperparameters for evaluation. Note that we did not find the official codes for [3] and [4] for the real-world experiment, so we reproduce the methods according to their papers. Aligning with the original experimental setting, we experiment on CIFAR-10 with four frameworks: DCGAN, WGAN, WGANGP and SNGAN. Inception Score and Fréchet Inception Distance are adopted as the evaluation metrics. The experimental results are presented as follows:
>
> Evaluation over Inception Score:
>
> |        | vanilla | Evolgan (a=0, b=80) | Evolgan (a=0.5, b=80) | Evolgan (a=0.5, b=20) | Evolgan (a=1, b=20) | DDLS  | AdvlatGAN-z |
> | ------ | ------- | ------------------- | --------------------- | --------------------- | ------------------- | ----- | ----------- |
> | DCGAN  | 5.848   | 5.815               | 5.887                 | 5.825                 | 5.791               | 5.502 | **6.086**       |
> | WGAN   | 6.590   | 6.528               | 6.496                 | 6.521                 | 6.546               | 2.607 | **7.334**       |
> | WGANGP | 7.295   | 7.349               | 7.316                 | 7.441                 | 7.236               | 7.074 | **8.279**       |
> | SNGAN  | 7.217   | 7.195               | 7.274                 | 7.298                 | 7.221               | 6.879 | **7.832**       |

---

> > ### Author Response · Authors · 2021-11-20
> > **Response to Reviewer ua7t (Cont.)**
> >
> > Evaluation over Fréchet Inception Distance:
> >
> > |        | vanilla | Evolgan (a=0, b=80) | Evolgan (a=0.5, b=80) | Evolgan (a=0.5, b=20) | Evolgan (a=1, b=20) | DDLS    | AdvlatGAN-z |
> > | ------ | ------- | ------------------- | --------------------- | --------------------- | ------------------- | ------- | ----------- |
> > | DCGAN  | 48.799  | 49.237              | 48.428                | 49.080                | 48.945              | 58.133  | **45.763**      |
> > | WGAN   | 34.439  | 34.685              | 34.650                | 34.564                | 34.565              | 221.367 | **32.854**      |
> > | WGANGP | 26.926  | 27.004              | 26.252                | 26.497                | 26.492              | 32.516  | **22.870**      |
> > | SNGAN  | 27.662  | 27.661              | 27.418                | 27.542                | 27.076              | 33.462  | **23.531**      |
> >
> > We have tuned multiple sets of parameters for Evolgan and DDLS and have presented the best results, which are still not very competitive, probably because there may exist some specific implementation techniques since we did not find the official implementations. The weak experimental improvement of Evolgan may stem from the dataset differences between Koncept512 training set and the evaluated set, which makes Koncept512 fail to provide reliable guidance (as mentioned in Appendix C.1). The quantitative results exhibit the effectiveness of AdvlatGAN-z.
> >
> >
> > **Q3: Does the presented method actually address the issues highlighted in Fig. 3?**
> >
> > **R3:** The issue in Fig. 3 is addressed by AdvLatGAN-z. As the original paper version does not cover experiments with the same setting as Fig. 3, we add additional experiments to show the effectiveness of AdvLatGAN-z. Note that AdvLatGAN-z works at the sample level (or distribution level as a distribution shift characterized by sample transforms) in latent space. In the Fig. 3 setting, when the sampling encounters invalid examples, AdvLatGAN-z is able to transform the bad $z$ to $z^*(z)$ with a better ability to avoid bad generation. Here we verify whether invalid samples can be improved with AdvLatGAN-z.
> >
> > We present the results in Fig. 9. The first row are bad generations (the first three correspond to those in Fig. 3) with the defects as follows: **first column**: the cat misses an ear; **second column**: white spots on the nose; **third column**: two bodies share one head; **fourth column**: green face; **fifth and sixth columns**: semi-existing glasses. The second row are results under AdvLatGAN-z. The results show that AdvLatGAN-z can effectively fix or avoid image defects. As you may notice there appears unexpected background variation in the third column, we believe this may stem from that the defect of the image is not significant enough for the discriminator, making the optimization less on target.
> >
> > We hope this response could help address your concerns, and wish to receive your further feedback soon.
> >
> > ---
> >
> > **References:**
> >
> > [1] Don't let your Discriminator be fooled. ICLR 2018.
> >
> > [2] Adversarial symmetric GANs: Bridging adversarial samples and adversarial networks. Neural Networks 2021.
> >
> > [3] Evolgan: Evolutionary generative adversarial networks. ACCV 2020.
> >
> > [4] Your GAN is secretly an energy-based model and you should use discriminator driven latent sampling. arXiv 2020.

---

> ### Author Response · Authors · 2021-12-03
> **Response to Additional Comment**
>
> Thank you for your further comment. It seems that your remaining concerns lie in the missing of some results in the paper and how the problem illustrated in Fig. 3 is addressed by our method. Here we make further clarification to these points.
>
> 1. Note that in Appendix D (see the experimental results in Table 9-12), we have already included the comparison with other "GANs + adversarial learning" methods and "GANs + latent exploration" methods in the paper. Besides, if you are talking about more baselines in CIFAR-10 case suggested by Reviewer JVu6, we have included our additional results in Table 4-5. For Fig. 3 setting, we have included the results in Sec. 4.2 (Results on AFHQ and FFHQ). Fig. 9 presents the visual results.
>
> 2. We explain how the Fig.3 problem is addressed more carefully. Fig.3 is started from equidistant sampling in latent space and the continuous generative mapping (DNN) guarantees the continuity in pixel space, which can cover invalid samples in the between of the manifolds, thus bad quality sampling is unavoidable with one DNN-based generator. In this paper, AdvLatGAN-z improves the latent space sampling to address the problem, which is able to transform the bad $z$ to $z^*(z)$ with a better ability to avoid bad generation. This helps to fix invalid samples in Fig. 3 to solve the problem. We verify whether invalid samples can be improved with AdvLatGAN-z and present the results in Fig. 9. Note here that AdvLatGAN-z would no longer keep the latent sampling equidistant, the former valid samples do not need to be presented repeatedly here. To illustrate with an example, in Fig. 3's second row, when the sampling encounters invalid examples (white spots on the nose), AdvLatGAN-z transforms the sampling to fix it (see Fig. 9's second column).
>
> We hope this response could help ease your concerns, and as we believe our work can contribute to this area, we sincerely hope that you could reconsider your rating.

---

### Official Review · Reviewer_NCez · 2021-11-04

**Correctness:** 3
**Technical Novelty And Significance:** 2
**Empirical Novelty And Significance:** 2
**Recommendation:** 5
**Confidence:** 4

**Main Review:**

This paper introduces their motivation convincingly with some easily comprehensible figures. However, the experiments to show the effectiveness of the proposed method are basically performed only with small scale datasets, so that it makes difficult to figure out how complex manifold in the target space can be handled by this approach, in particular, it is unclear how it can scale to the ImageNet dataset. The discussion to compare the proposed method with other work seems to be not enough because there are some more research that shed light on the importance of latent vector transformation. For example, the series of StyleGAN work is continuously improving the performance by precisely analyzing the relationship between the latent space and the pixel space. They also firstly adopted to transform the latent vector z to another latent space W using a mapping network. That should be discussed as one of the possible approach to transform the latent vector, i.e., using a trainable function instead of I-FGSM.

Some figures to show the effectiveness of the proposed method seems difficult to interpret as the captions claim, e.g., Figure 5. It was difficult to find the clear difference between the results of "Anti-diverse" and "Original", and the "Diverse" results look worse than other two in terms of the quality (if it is intended because the figure tries to show the increase of diversity, but it still unclear that the "Diverse" row in the Figure 5 has higher diversity compared to other two.)

**Summary Of The Paper:**

This paper applies two existing techniques derived from adversarial example literature and MSGAN to improve the quality and diversity of generated samples by GAN. The former idea from [Goodellow et al., 2014b] is used to shift a latent vector z originally sampled from the Gaussian, which is different from the original adversarial attack paper that transforms the input image. The direction to move the latent vector is calculated by using I-FGSM with a standard loss for the generator proposed in the vanilla GAN paper. The experiments show that this technique can improve the quality of generated samples. Furthermore, this paper also consider how to improve the diversity of generated samples by transforming the latent vectors before putting them into the generator network. The approach for that is based on MSGAN's idea that mines hard latent vectors for the discriminator. The authors combine these two techniques and achieved better results compared to DCGAN, WGAN, WGAN-GP, and SNGAN models on several relatively small scale datasets such as CIFAR-10, STL-10, etc.

**Summary Of The Review:**

The paper is well organized and easy to follow the authors' claims but the qualitative comparison results seems difficult to be interpret as the captions state. And from the perspective of transformation of the latent vector z, there can be more discussion in the comparison with other methods such as StyleGAN that transform z before putting it into the generator.

---

> ### Author Response · Authors · 2021-11-20
> **Response to Reviewer NCez**
>
> Thank you for the detailed feedback. Here are our responses to your comments:
>
> **Q1: Discussion about the latent transform in StyleGAN.**
>
> **R1:** Thanks for pointing out our missing related work on latent space transform, and we will integrate StyleGAN [1] into our related work section. StyleGAN does introduce a latent space transform by using a mapping network, but we believe that the purpose of its transform significantly differs from ours. Stylegan's latent transform aims at better decoupling the various styles, and to some extent eliminating the limitation from the training data (see Fig. 6 in [1]) to achieve this goal. In contrast, our transform in latent space is designed to overcome the difficulty of fitting continuous latent distribution to the real distribution which is supported on disjoint manifolds, not to eliminate training data's bad effect on decoupling. Instead, it leverages real data's distribution structure to adjust the latent distribution.
>
> StyleGAN's mapping network is indeed an enlightening point in terms of latent transform implementation. Actually, we have considered using trainable networks to achieve the transform, and what made us discard this idea is the need to introduce this technique into training, as introducing training new networks can increase the computational burden. Also, we believe that for a well-trained model, the changes to the latent vectors required for overcoming generation flaws shall be small and are more reasonably interpreted as perturbations. The adversarial attack and adversarial training methods can be a better match for the preceding scenarios.
>
> **Q2: Poor interpretability of the Fig. 5 caption claim.**
>
> **R2:** We are sorry for not providing a clear enough description to Fig. 5 in the caption, and we have improved the figure and rephrased the caption as follows:
>
> Each of the six columns shows two compared pairs of generated example (first pair: 1st + 2nd row; second pair: 3rd row + 2nd row) by our diversity driven iterative transform scheme in latent space with the same number of iterations to obtain $z^*$ from initial $z$. **Middle row:** generation by vanilla latent sampling $z$; **Top:** by latent sample transformed by Eq. 8; **Bottom:** by latent sample transformed by Eq. 8's inverse form. It shows using Eq. 8 generates more similar image pairs serving as hard samples for training, rather than using its inverse form (see quantitative results in Table 1).
>
> We hope this response could help address your concerns, and wish to receive your further feedback soon.
>
>
> ---
> **References:**
>
> [1] A style-based generator architecture for generative adversarial networks. CVPR 2019.

---

> ### Author Response · Authors · 2021-12-03
> **Further Response by Authors**
>
> Thanks for your comments and suggestions which have inspired us a lot to improve the paper. For the issues raised above, we have discussed StyleGAN and included it in the related work section, and we have restated Fig. 5 and Fig. 8's captions with more clear descriptions. We are sorry that due to the limited time, we may not provide the results for large-scale datasets' results for proposed training algorithms. But for AdvlatGAN-z, we have supplemented the experiment on AFHQ and FFHQ to evaluate its effectiveness on larger data (see Sec. 4.2 and Fig. 9).
>
> We believe that our current PDF version is further improved in experiment design with significant performance gain (at most 12%~47% advantage on different metrics and settings). We believe our work can contribute to this area and we sincerely hope that you could reconsider your rating.

---

### Author Response · Authors · 2021-11-20
**General Response by Authors**

Dear area chair and reviewers,

We appreciate the reviewers' time and valuable comments and we are sorry for the late response as we have been working on additional experiments as suggested by the reviewers. Overall, the reviewers considered our work to be well organized and easy to follow (NCez, ua7t, uSqk), with clear figures or logic (NCez, uSqk), refined in ablation study experiment design (uSqk), and the idea was commended as elegant, clear or interesting (ua7t, uSqk). The major concerns lie in inadequate discussion or comparison with methods interplaying between GANs and adversarial attacks or latent space exploration (NCez, ua7t, JVu6), lack of large-scale evaluation (NCez, ua7t, JVu6), whether or how the problem in Fig.3 is solved (ua7t, uSqk), problems or improvement suggestions for existing experiments (uSqk, JVu6) and poor interpretability of some captions (NCez, uSqk).

We appreciate and acknowledge most of the comments, which are valuable for us to refine our work, and for other issues that require clarification, we provide detailed answers point by point in the following individual responses. We believe that the main shortcomings of our work lie in the inadequacy of the experiments and the related work discussion, which makes it our main focus during the rebuttal period.

We summarize our work in the rebuttal phase as follows:

1. We compare our techniques with other methods interplaying between GANs and adversarial learning. We reproduce [1] and [2] for comparison, which are respectively related to perturbations on the generated samples and the real samples.
2. We compare our techniques with other methods interplaying between GANs and latent space exploration. We reproduce and compare representative latent exploration works [3] and [4] with AdvLatGAN-z.
3. We discuss and include more works on latent space mining in GAN including StyleGAN [5], LOGAN [6], ClusterGAN [7] and [8] as mentioned by reviewers NCez and JVu6 in the main paper.
4. We present a further explanation for the role of the latent transform in Fig. 1, along with experiments on Fig. 3 setting to further illustrate methodological effectiveness.
5. We re-evaluate the experimental results with the density/coverage metric [9] which is capable of separating the fidelity evaluation and the diversity evaluation.
6. We supplement more baselines LSGAN [10], WGAN-div [11] and ACGAN [12] for comparison in CIFAR-10 evaluation setting, along with LSGAN and WGAN-div in STL-10 evaluation setting (ACGAN does not work on unlabeled STL-10).
7. We restate some of the figure captions with more clear descriptions.
8. We complete the discussion on other correlated issues in individual responses.

The efforts we make have already been added to the paper.

Beyond these, we are working on experiments with large-scale datasets, but due to the experiment scale and limited time, the experiments are still in progress. We will update the results as soon as they are available.

We hope that our works can ease the reviewers' concerns and further strengthen our contributions.

Your further feedback is welcomed and appreciated.


---
**References:**

[1] Don't let your Discriminator be fooled. ICLR 2018.

[2] Adversarial symmetric GANs: Bridging adversarial samples and adversarial networks. Neural Networks 2021.

[3] Evolgan: Evolutionary generative adversarial networks. ACCV 2020.

[4] Your GAN is secretly an energy-based model and you should use discriminator driven latent sampling. arXiv 2020.

[5] A style-based generator architecture for generative adversarial networks. CVPR 2019.

[6] Latent optimisation for generative adversarial networks. arXiv 2019.

[7] Clustergan: Latent space clustering in generative adversarial networks. AAAI 2019.

[8] Effect of the Latent Structure on Clustering with GANs. IEEE Signal Processing Letters 2020.

[9] Reliable fidelity and diversity metrics for generative models. ICML 2020.

[10] Least squares generative adversarial networks. ICCV 2017.

[11] Wasserstein divergence for gans. ECCV 2018.

[12] Conditional image synthesis with auxiliary classifier gans. ICML 2017.

---

### Author Response · Authors · 2021-11-26
**Look Forward to Your Reply to Our Rebuttal**

Dear reviewers, thanks for your comments and suggestions which have inspired us a lot to improve the paper. We are sincerely looking forward to your reply and we could provide more information if needed.

---

### Author Response · Authors · 2021-11-30
**Look Forward to Your Feedback**

Dear reviewers, since the discussion phase is approaching the end, would you please provide some feedback?

---

### Decision · Program_Chairs · 2022-01-20

**Decision:**

Reject

**Comment:**

To improve the generative adversarial nets, the paper proposes to add an implicit transformation of the Gaussian latent variables before the top-down generator. To further obtain better generations with respect to quality and diversity, this paper introduces targeted latent transforms into a bi-level optimization of GAN.  Experiments are conducted to verify the effectiveness of the proposed method. The paper is highly motivated and well-written, but the experiment part still needs to be strengthened because the goal of the paper is to improve the GAN training, comprehensive and thorough evaluation of the proposed method is necessary.

After the first round of review, in addition to the clarification issue and missing reference issue, two reviewers point out that the method is only tested in small-scale datasets, and suggest authors evaluate the performance of the proposed method in more complex datasets. Two reviewers point out that the experimental validation and comparison to prior approaches are insufficient. During the rebuttal, the authors provide extra experiment results to partially address some issues.  However, most of the major concerns from other reviewers, such as (i) how are the performance of the method in large scale datasets that have complex latent space manifolds, (ii) non-convincing performance gain, and unclear problem setup, still remain. After an internal discussion, AC agrees with all reviewers that the current paper is not ready for publication, thus recommending rejecting the paper. AC urges the authors to improve their paper by taking into account all the suggestions provided by the reviewers, and then resubmit it to the next venue.